# Secondary structures that regulate mRNA translation provide insights for ASO-mediated modulation of cardiac hypertrophy

Omar M. Hedaya [1,2], Kadiam C. Venkata Subbaiah[1], Feng Jiang [1,2], Li Huitong Xie [3], Jiangbin Wu [1], Eng-Soon Khor [1], Mingyi Zhu [2,4], David H. Mathews [2,4,5], Chris Proschel[3] & Peng Yao [1,2,4,5] ✉

Translation of upstream open reading frames (uORFs) typically abrogates translation of main (m)ORFs. The molecular mechanism of uORF regulation in cells is not well understood. Here, we data-mined human and mouse heart ribosome profiling analyses and identified a double-stranded RNA (dsRNA) structure within the *GATA4* uORF that cooperates with the start codon to augment uORF translation and inhibits mORF translation. A trans-acting RNA helicase DDX3X inhibits the GATA4 uORF-dsRNA activity and modulates the translational balance of uORF and mORF. Antisense oligonucleotides (ASOs) that disrupt this dsRNA structure promote mORF translation, while ASOs that base-pair immediately downstream (i.e., forming a bimolecular double-stranded region) of either the uORF or mORF start codon enhance uORF or mORF translation, respectively. Human cardiomyocytes and mice treated with a uORF-enhancing ASO showed reduced cardiac GATA4 protein levels and increased resistance to cardiomyocyte hypertrophy. We further show the broad utility of uORF-dsRNA- or mORF-targeting ASO to regulate mORF translation for other mRNAs. This work demonstrates that the uORF-dsRNA element regulates the translation of multiple mRNAs as a generalizable translational control mechanism. Moreover, we develop a valuable strategy to alter protein expression and cellular phenotypes by targeting or generating dsRNA downstream of a uORF or mORF start codon.

The translation of messenger RNAs (mRNAs) is a fundamental process required to produce proteins that in turn conduct biochemical reactions and other biological functions in all organisms. This process is tightly regulated to control protein levels in response to physiological and pathological cues[1,2]. During canonical translation initiation, the 40S small ribosomal subunit is recruited by eukaryotic translation initiation factor 4F (eIF4F; also known as eIF4E-eIF4G-eIF4A trimeric complex) to form the 43S preinitiation complex (PIC). The PIC then

[1]Aab Cardiovascular Research Institute, Department of Medicine, University of Rochester School of Medicine & Dentistry, Rochester, NY 14642, USA. [2]Department of Biochemistry & Biophysics, University of Rochester School of Medicine & Dentistry, Rochester, NY 14642, USA. [3]Department of Biomedical Genetics, University of Rochester School of Medicine & Dentistry, Rochester, NY 14642, USA. [4]The Center for RNA Biology, University of Rochester School of Medicine & Dentistry, Rochester, NY 14642, USA. [5]The Center for Biomedical Informatics, University of Rochester School of Medicine & Dentistry, Rochester, NY 14642, USA. ✉e-mail: peng_yao@urmc.rochester.edu

scans the 5′-untranslated region (5′ UTR) of the mRNA towards the coding sequence start codon, and translation initiation occurs. While scanning, the PIC interacts with cis-acting elements that regulate mRNA translation[3]. One of the most common elements is upstream open reading frames (uORFs) that encode short peptides and are present in approximately 50% of human mRNAs[4]. Translating these peptide-encoding sequences leads to suppression of the main open reading frame (mORF; also referred to as the protein-coding sequence, CDS) because the PIC is consumed during translation initiation in uORFs[5,6]. Despite this seemingly wasteful process, uORFs have persisted throughout evolution, suggesting this mechanism is vital to protein expression and physiological homeostasis. uORFs are known to regulate the expression of proteins that play important roles in diverse biological processes such as cell differentiation or catabolic metabolism, and may contribute to disease processes[7–10]. However, considering the large number of uORFs and the great diversity of 5′ UTRs in the human transcriptome, the mechanistic regulation and functional significance of most uORFs remain unknown.

RNA secondary structures formed by intramolecular base-pairing constitute another frequently present cis-acting RNA element that can either activate or inhibit mRNA translation, as exemplified by internal ribosome entry sites or ribosome blockades, respectively[3]. Existing evidence suggests that double-stranded (ds)RNA structures and uORFs could engage in crosstalk. A recent study using *Saccharomyces cerevisiae* showed that RNA structures in mRNA 5′ UTRs activate translation initiation at upstream cognate or non-cognate start codons[11]. It remains unclear whether the same mechanism is present in human cells. Moreover, human 5′ UTRs, compared to yeast 5′ UTRs, manifest greater complexity and diversity. Although 5′ UTR dsRNA structures often repress mRNA translation, mRNA 5′ UTRs paradoxically became longer and more structured with the presence of dsRNA stem-loops throughout evolution[3]. Therefore, some human uORFs and RNA structures may be positioned in a way that is conducive to uORF translation. Hints that dsRNA-mediated regulatory mechanisms indeed exist in human cells emerged from recent findings showing that stem-loops formed by the pathogenic expansion of CGG or GGGGCC repeats in the *FMR1* mRNA 5′ UTR or *C9ORF72* pre-mRNA intron 1 activate translation initiation at non-AUG codons immediately upstream or within the dsRNA stem loops[12,13]. Despite prior studies demonstrating the importance of cis-residing uORFs and dsRNA structures in individually regulating mRNA translation, their combined molecular action on mRNA translation and their impact on human cell physiology remain largely unknown. Understanding their interplay has the potential to develop RNA-targeting therapeutics that include antisense oligonucleotides (ASOs) to treat human diseases[14–16].

Here we examine the interplay of uORF and dsRNA structures in the 5′ UTR of the transcription factor GATA-binding protein 4 (GATA4) mRNA that encodes a zinc finger transcription factor (TF) as a central regulator of cardiomyocyte development and hypertrophy[17]. In this study, we use *GATA4* 5′ UTR-based reporter mRNAs to demonstrate that a dsRNA structure situated downstream of the *GATA4* uORF initiation codon can enhance uORF translation and, thereby, repress mORF translation. Moreover, disrupting the dsRNA element diminishes uORF-mediated suppression and promotes mORF translation. Based on this uORF-dsRNA regulatory mechanism, we develop two types of uORF-targeting ASOs to either inhibit or activate uORFs, thereby promoting or inhibiting mORF translation of *GATA4* mRNA. We show that genetic and ASO-mediated inactivation of uORF translation increases *GATA4* mRNA translation in human CMs to control cellular hypertrophy. The Gata4 uORF-activating ASO also effectively antagonizes CM hypertrophy in mice under cardiac stress caused by neurohumoral stimulation and transverse aortic constriction. Lastly, we optimized and applied the uORF-targeting ASOs for additional mRNAs beyond *GATA4* and repurposed some of the ASOs to target the mORF and increase translation directly.

## Results

### dsRNA structure downstream of initiation codon activates uORF translation

Among the 89,306 human 5′ UTR sequences annotated in the Ensembl Genome Browser, we found that 54.44% contained at least one uORF (Fig. 1a and Supplementary Data 1). Among 108,652 mRNA transcripts annotated, 5′ UTRs contained a median of 61.1% of G or C nucleotides (interquartile range [IQR]: 52.5–69.4%). In contrast, medians for the coding sequence (CDS) and 3′ UTR are 51.7% (IQR: 45.1–58.7%) and 44.1% (IQR: 36.8–53.3%), respectively (Fig. 1b and Supplementary Data 1). High GC content naturally results in a higher degree of RNA structure[18]. Together, this data indicates that both uORFs and RNA secondary structures are enriched in human mRNA 5′ UTRs.

uORFs are well known to inhibit mORF translation[7–10,19], while double-stranded RNA (dsRNA) structures embedded in 5′ UTR can inhibit or activate translation depending on their location and structural features[3]. Nevertheless, it is unclear how dsRNA-uORF sequences crosstalk to alter mORF translation. To answer this question, we generated reporter mRNAs that would examine the interaction between an upstream uORF and a downstream dsRNA hairpin on the expression of a reporter mORF (Fig. 1c). Specifically, a 5′ UTR was followed by an artificial uORF that was interrupted immediately downstream of its AUG start codon with varying lengths (0-to–20 nucleotides) of an unstructured CA repeat and upstream of a well-established stem-loop hairpin (hp) KanHP1[20]. This was followed by a 27-nucleotide spacer, and then an mORF that encodes Firefly luciferase (FLuc). A uORF-free control reporter (counterpart of the −2 AUG-hp reporter) was generated by mutating the uORF AUG start codon to a UUG codon. Each reporter plasmid (Fig. 1c, left, Supplementary Fig. 1a, and see sequences in Supplemental Information)[20] was transiently introduced into HEK293T cells with a reference plasmid encoding Renilla Luc (RLuc). FLuc activities were normalized to RLuc activities, the latter of which controlled for variations in transfection efficiencies and cell-extract recoveries.

Results revealed that FLuc activity was most effectively decreased (i.e., to $41.5 \pm 3.1\%$ or $49.2 \pm 6.1\%$ of its uORF-lacking counterpart, respectively) when the uORF start codon resided 2 or 5 nucleotides upstream of the hp, hereafter referred to as the '−2 reporter' or '-5 reporter' because the hp is inserted at the G of the AUG (and the G is defined as 0) (Fig. 1c, right). FLuc activity was modestly decreased when the uORF start codon resided 8 nucleotides upstream of the hp (i.e., to $68.6 \pm 13.1\%$ of its uORF-lacking counterpart), and was unaffected when the uORF start codon resided 11 through 23 nucleotides upstream of the start codon (Fig. 1c, right). Notably, normalized reporter mRNA levels, determined using RT-qPCR, were unchanged between the uORF-containing mRNA and the non-uORF-containing control mRNA (Fig. 1d), indicating that the observed changes in FLuc activity resulted from alterations in mRNA translation.

To elucidate the role of the hairpin structure on uORF activity, we introduced mismatches that disrupted the hp in the −2 reporter, generating −2 wild-type (WT) hp (folding free energy $\Delta G = -12.4$ kcal/mol) and −2 mismatch (MM) hp (folding free energy $\Delta G = -3.9$ kcal/mol) (Fig. 1e, left). The −2 MM hp reporter manifested no hairpin-mediated inhibition of FLuc mORF translation since it produced FLuc activity comparable to the −2 WT reporter (Fig. 1e, right) from the same level of mRNA (Fig. 1f). In contrast, these mismatches produced limited changes in FLuc activity produced by the non-uORF-containing (No AUG) reporter (Fig. 1e). We conclude that combining a uORF AUG start codon with an optimal distance of 2-to-5 nucleotides upstream of a stable dsRNA hp most effectively represses mORF translational activity.

To explore the mechanism underlying the dsRNA-mediated influence of uORF translation on mORF translation, we examined

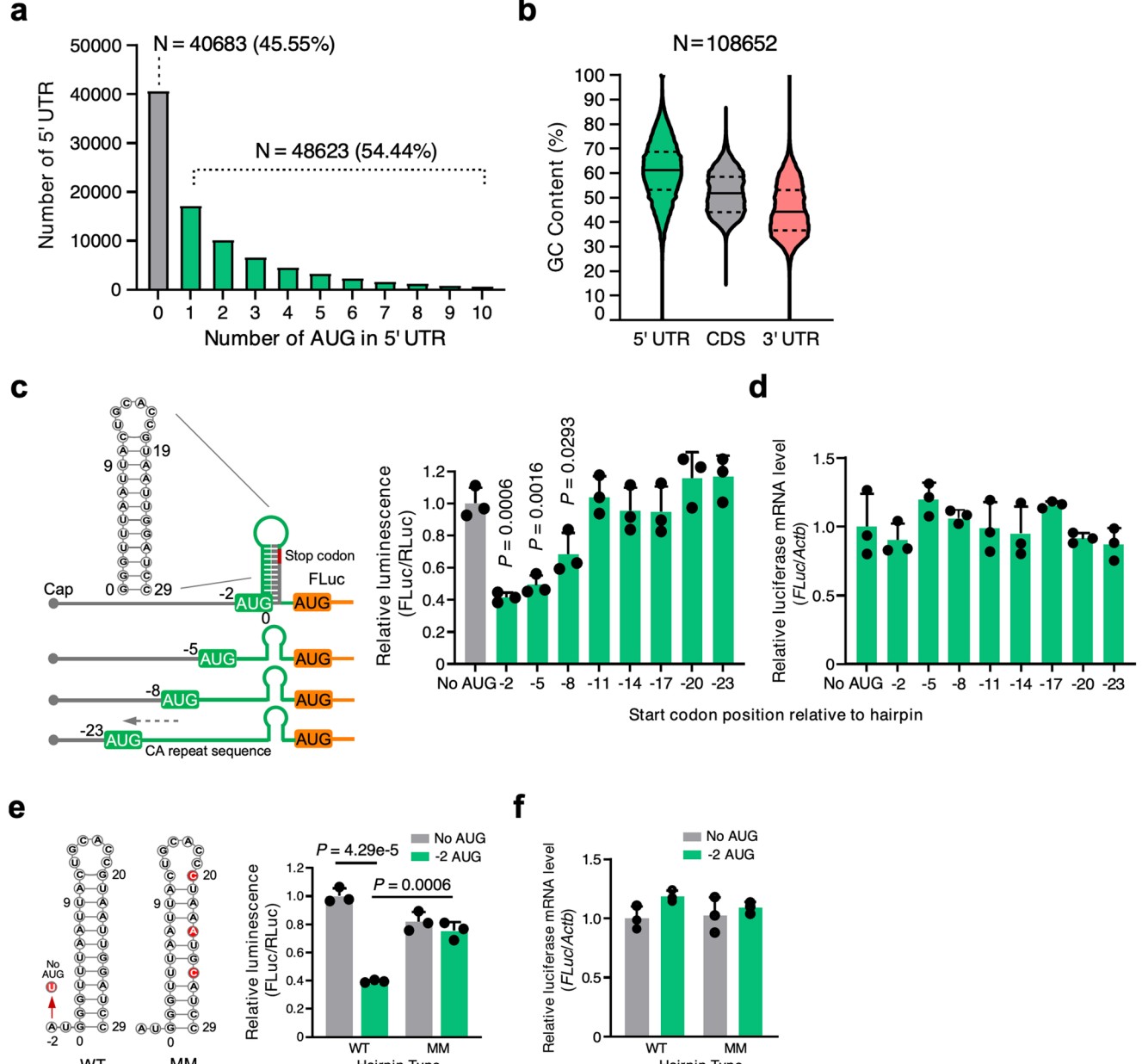

**Fig. 1 | Crosstalk between a uORF and an adjacent double-stranded RNA structural element inhibits mORF translation. a**, **b** Sequence analysis of Ensemble transcripts demonstrates the abundance of 5′ UTR start codons (**a**) and 5′ UTR GC content relative to the CDS or 3′ UTR (**b**). The solid line indicates the median, while the dotted lines indicate the interquartile range. More than half the human mRNA 5′ UTRs contain AUG start codons. **c** Left panel: schematic of FLuc reporter constructs. Right panel: dual luciferase reporter assay using a series of constructs that contain uORF start codon and adjacent dsRNA KanHP1 located at increasing distances in 3 nt intervals. **d** RT-qPCR of *FLuc* mRNA normalized to *ACTB*

from cells in (**c**). No AUG control: ATG-to-TTG mutation in the −2 construct. **e** Dual luciferase reporter assay using mutant constructs. No AUG: ATG-to-TTG mutation. AUG −2: start codon is located at the −2 position relative to the hairpin (sequences available in Supplementary Information). WT: stable hairpin. MM: three mismatched mutations were introduced in the hairpin to disrupt the dsRNA structure. **f** RT-qPCR of *FLuc* mRNA normalized to *ACTB* from cells in **e**. Data are represented as mean ± SD. **$P < 0.01$, ***$P < 0.001$; Statistical significance was confirmed by unpaired two-tailed Student *t* test for **c**–**f** (*N* = 3 biological replicates). Source data are provided as a Source Data file.

the effect of the hp on translation initiation. We used four [32P]-labeled in vitro-transcribed mRNAs: a linear mRNA that consisted of CA repeats; ORF mRNA encoding a 7-amino acid peptide; hp mRNA lacking an ORF; and −2 ORF hp mRNA (where the hp is inserted at the G of the AUG codon). All of these mRNAs also contained a poly(A) tail. Each 32P-labeled mRNA was incubated in translationally active rabbit reticulocyte lysates (RRLs) (Supplementary Fig. 1b, c). Lysates were then fractionated by ultracentrifugation using 10–35% sucrose gradients, and mRNA was detected using liquid scintillation. Results demonstrated that linear mRNA and ORF mRNA co-

sedimented with free mRNP (Supplementary Fig. 1d, in gray and in yellow), hp mRNA co-sedimented with 40S ribosome subunits (Supplementary Fig. 1d, in pink), and the -2 ORF hp mRNA co-sedimented with 80S ribosome (Supplementary Fig. 1d, in green), suggesting that the hp promotes translation initiation at the AUG start codon. This result is consistent with an earlier study[21] (Supplementary Fig. 1d, in pink). Taken together, our results indicate that a hp immediately downstream of a uORF start codon cooperates to inhibit mORF translation in a mechanism that is abolished when the hp is destabilized.

## Identification of translationally active uORFs in mRNAs encoding biologically relevant proteins

Several studies have surveyed uORFs at a transcriptome-wide level using ribosome profiling (Ribo-seq)[14,22]. Relevant to our interest in cardiac biology, we mined previous Ribo-seq datasets derived from human or mouse hearts[14,22] to identify translationally active uORFs (Supplementary Fig. 2a, left). Gene ontology analysis[23] of genes containing a translationally active uORF (Supplementary Fig. 2a, middle) revealed 49 transcription/chromatin regulators, including transcription factors (TFs), as the most-enriched gene set. These genes include the TF *GATA4*, *TBX5*, *MYOCD*, and *NKX2-5* (Supplementary Fig. 2a, right; Supplementary Data 2), which are well-established regulators of cardiac development and cardiomyocyte (CM) growth. In addition, a significant number of uORF-bearing transcripts belong to RNA-binding proteins, translation factors, and metabolic enzymes (Supplementary Data 2).

## A dsRNA element is located downstream of the uORF in *GATA4* mRNA

We focused further studies on the function of the GATA4 uORF because of its importance to cardiac biology[24] and its simplicity, consisting of a single uORF across multiple mammals (Supplementary Fig. 2b) such that the uORF AUG is followed by a dsRNA region (as predicted using RNAstructure)[25] (Supplementary Fig. 2c, upper left) present across multiple species. Also, the fact that GATA4 is an essential transcription factor required for CM growth and hypertrophy during cardiac development and stress[24,26,27] offers an opportunity to relate our biochemical studies to cellular and animal phenotypes regarding the cardiovascular system[27–31].

To confirm the predicted dsRNA region, we employed in vitro RNA selective 2′ hydroxyl acylation analyzed by primer extension (SHAPE) to determine the secondary structure of the putative hp of *GATA4* mRNA 5′ UTR. To this end, we in vitro-transcribed a human *GATA4* mRNA 5′ UTR fragment that consisted of nucleotides 209–465 and contained the full dsRNA region and intact uORF (Fig. 2a). Following SHAPE and quantification using SAFA[32,33], we obtained and normalized the SHAPE reactivity values of each nucleotide (Fig. 2b, c)[34]. Two independent replicates of the SHAPE assays strongly agreed with each other, indicating the reproducibility of this analysis (Spearman $R^2 = 0.95$; Supplementary Fig. 2d). The resulting SHAPE values were applied as folding constraints to predict RNA structure using RNAstructure. The structure derived supports the presence of a 10-nt double-stranded stem starting at $G^{384}$ of the uORF start codon (Fig. 2d, upper panel) and agrees with the predicted local structure of *GATA4* mRNA (Fig. 2d, lower and right panels).

## dsRNA downstream of an uORF start codon promotes uORF-mediated inhibition of the mORF translation in *GATA4* mRNA

We next investigated how a biologically relevant uORF-dsRNA element functions in translational control of an associated mORF using HEK293T cells. To this end, we generated plasmids encoding reporter mRNAs that consisted of the *GATA4* 5′ UTR, including the uORF and dsRNA that begins with the G of the uORF AUG start codon, fused to FLuc. Reporter mRNAs harbored either the WT or a MM dsRNA stem sequence that destabilizes the dsRNA (Supplementary Fig. 2c, upper). This was confirmed using in vitro SHAPE (Supplementary Fig. 2c, lower). We also generated plasmids encoding reporter mRNAs containing either the WT uORF AUG or an AUG-to-UUG start codon mutation (ΔuORF) with or without the MM mutations (Fig. 2e, left. After transient expression in HEK293T cells, all ΔuORF reporters showed the same FLuc activity irrespective of the presence of the MM mutations (Fig. 2e, middle). For the ΔuORF reporters, the relative FLuc activity was 1.57 ± 0.10-fold enhanced over that of the WT uORF AUG, supporting the notion that uORF translation inhibits mORF translation. The inhibition of mORF translation was nullified by the MM mutations

and was then recovered with the introduction of compensatory mutations that restored base-pairing in the MM stem (Fig. 2e, middle). Importantly, WT and mutated FLuc reporters produce comparable mRNA levels (Fig. 2e, right). To confirm the translation of the uORF-encoded peptide, we inserted a 3x FLAG tag at the N- and C-terminus of the *GATA4* uORF in the WT and the ΔuORF FLuc reporter constructs. As expected, the ΔuORF reporter showed higher FLuc activity than the WT reporter, suggesting a higher mORF protein expression (Supplementary Fig. 2e, left), while the expression of FLAG-tagged uORF peptide was abolished for the ΔuORF reporter (Supplementary Fig. 2e, right).

We next examined whether the mechanism of uORF-dsRNA-mediated regulation of mORF translation is generalizable to other protein-coding mRNAs. Since low Shannon entropies identify mRNAs likely to fold into a single secondary structure[35], we calculated the Shannon entropies for sequences upstream of the mORF of mRNAs that contain translated uORFs in the human heart[14]. Two mRNAs encoding critical mitochondrial proteins, *MFN1* (Mitofusin 1)[36] and *DHTKD1* (dehydrogenase E1 and transketolase domain containing 1)[37], show low Shannon entropies among the lowest (2.5 percentile) of those we estimated (Supplementary Fig. 2f and Supplementary Data 3), suggesting a low probability of forming multiple alternative secondary structures. *MFN1* and *DHTKD1* mRNAs also contain a single uORF with a putative downstream dsRNA element (Supplementary Fig. 2g). MM mutations were introduced in the predicted dsRNA structures of both WT and ΔuORF MFN1-FLuc and DHTKD1-FLuc plasmids (Supplementary Fig. 2g). After expression in HEK293T cells with the RLuc reference plasmid, mRNAs harboring the MM mutations partially abrogate the uORF-mediated inhibition of FLuc activity (Supplementary Fig. 2h). These results demonstrate that a dsRNA element can cooperate with a uORF initiation codon to activate uORF translation, thereby inhibiting mORF translation in multiple biologically relevant mRNAs.

## RNA helicase DDX3X regulates uORF-dsRNA activity in *GATA4* mRNA

To elucidate the impact of trans-acting factors that might affect the activity of the GATA4 uORF, we tested the activity of the WT and ΔuORF GATA4 5′ UTR-bearing FLuc reporters following the depletion of DDX3X (DEAD-box helicase 3 X-linked)[11,38], eIF1 (eukaryotic translation initiation factor 1), eIF5 (eukaryotic translation initiation factor 5)[39], DENR (density regulated re-initiation and release factor)[40], and DHX29 (DExH-box helicase 29)[41] which were previously shown to regulate uORF activity of various mRNAs (Supplementary Fig. 2i, j). The knockdown of DDX3X specifically inhibited the activity of the wild-type FLuc reporter, while the mutant reporter with uORF deletion remained unaffected. In contrast, siRNA knockdown of multiple other uORF or dsRNA regulatory factors did not affect the activity of the wild-type FLuc reporter (Supplementary Fig. 2i, j). These results suggest that DDX3X might reduce the stringency of uORF start codons similarly as its homolog protein Ded1p acts in yeast[11].

As a helicase, this likely is due to the unwinding of the stable dsRNA structure proximal to the uORF. This hypothesis was tested using the same set of 5′ UTR reporters with mutations in the 10-bp stem and their ΔuORF counterpart in HEK293T cells transfected with a range of siRNA concentrations (Supplementary Fig. 2k). The MM mutant abolished its DDX3X dependence, confirming how essential dsRNA is for uORF-mediated inhibition of FLuc mORF translation based on a uORF-dsRNA synergy. The dependence of DDX3X was also restored for the rescue reporter (Fig. 2e, left panel), as indicated by reduced FLuc activity to the WT control reporter level (Supplementary Fig. 2k). Moreover, the interplay between dsRNA and uORF was further highlighted by the lack of DDX3X dependence of the ΔuORF variant.

We next asked the question of whether DDX3X is differentially regulated for modulating the uORF-dsRNA function upon stress conditions. A previous study reported a dose- and time-dependent

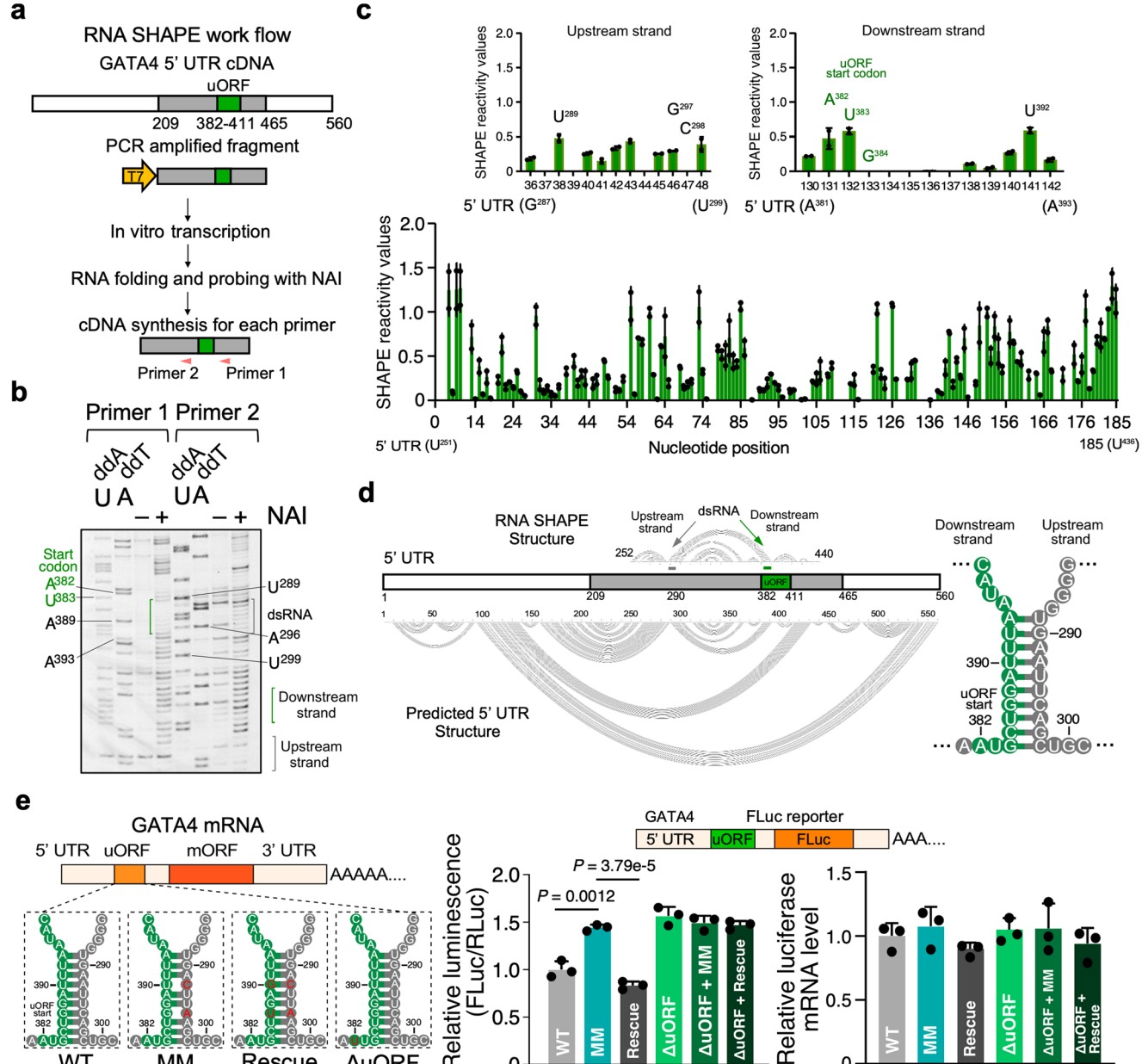

**Fig. 2 | Double-stranded RNA element cooperates with initiation codon to activate *GATA4* uORF translation. a** Schematic highlighting the key steps of RNA SHAPE workflow used for the human *GATA4* uORF-dsRNA-bearing RNA sequence. **b** RNA SHAPE analysis of human *GATA4* uORF-dsRNA-bearing RNA region. A sequencing gel resolved individual nucleotides of the SHAPE assay products. Primers 1 and 2 were used to detect the downstream and upstream dsRNA strands in the *GATA4* RNA. Experiments in **b** were carried out twice, and one representative result is shown. **c** Densitometric quantification of band intensity corresponding to the structured nature of a given nucleotide using SAFA. The upper inserts represent two strands of the dsRNA. Lower NAI SHAPE reactivity (band intensity) indicates dsRNA structure. Two biological replicates from **b** were plotted. **d** The predicted lowest free energy secondary structure of human *GATA4* uORF surrounding region based on SHAPE reactivity data using RNAstructure. The resolved region with detectable SHAPE activity is at 252–440 nt. **e** Left: Schematic of firefly luciferase (FLuc) reporters that include variants of the full-length RNA sequences of human *GATA4* mRNA before the start codon of mORF (including uORF-dsRNA). ΔuORF ATG-to-TTG start codon mutation, MM mismatch mutations in dsRNA, Rescue:mismatch and compensatory mutations. Middle: dual luciferase reporter assay with WT and mutants. Following transfection into HEK293T cells, FLuc levels were normalized to a control RLuc reporter. Right: RT-qPCR of *FLuc* mRNA normalized to *ACTB*. Data are represented as mean ± SD. **$P < 0.01$; Statistical significance was confirmed by an unpaired two-tailed Student $t$ test for **e** ($N = 3$ biological replicates). Source data are provided as a Source Data file.

induction of DDX3X protein expression by a bacterial toxin, lipopolysaccharide (LPS)[42], which we confirmed (Supplementary Fig. 2l). At these conditions, we compared the WT or MM, and with either a uORF or a ΔuORF mutation in HEK293T cells treated with LPS versus vehicle upon siRNA KD of DDX3X or mock KD. LPS-mediated stress increased FLuc reporter activity only in the WT reporters, while the MM and ΔuORF were resistant to this stimulation (Supplementary Fig. 2m). On the other hand, KD of DDX3X abolished this effect from LPS,

confirming the dependence of LPS-induced DDX3X in regulating mORF protein expression without changing the mRNA expression level (Supplementary Fig. 2n). Taken together, these findings suggest a role of DDX3X in regulating the dsRNA-uORF crosstalk in the GATA4 5′ UTR, which might be through RNA unwinding.

Intriguingly, this regulatory activity of DDX3X did not correlate with the ability of any of these reporter mRNAs to be pulled down by DDX3X in our RNA-binding protein immunoprecipitation (RIP) assay

(Supplementary Fig. 2o, p). One possible explanation is that RNA helicases are promiscuous and might bind to a given mRNA at multiple sites irrespective of the sequence or structure. Because the sequence difference between these reporters is only a few nucleotides, the change in binding affinity might be negligible.

### Design of uORF-targeting antisense oligonucleotides for GATA4 regulation

Our finding that a uORF-dsRNA configuration can activate uORF translation, which in turn inhibits mORF translation (Fig. 2 and Supplementary Fig. 2e), led us to test what effect antisense oligonucleotides (ASOs) that base-pair near the uORF AUG start codon of cellular *GATA4* mRNA may have on GATA4 protein production. To mimic the KanHP1 hp used in Fig. 1c, we screened a number of *GATA4*-specific 16-nucleotide ASOs that base-pair starting either (i) 3-nucleotides upstream of the uORF AUG start codon (uORF-ASO_−3), (ii) at the A of the AUG start codon (ASO_0), (iii) at the G of the AUG start codon (uORF- ASO_+2), or (iv) at 4-, 7-, or 10-nucleotides downstream of the G of the AUG start codon (uORF-ASO_+6, uORF-ASO_+9, uORF-ASO_+12) (Fig. 3a, left). We reasoned that these ASOs, which replace the endogenous dsRNA element with a longer and more stable intermolecular dsRNA element, would inhibit mORF translation. Indeed, results obtained using uORF-ASOs recapitulated those obtained using the artificial uORF reporter plasmids (Fig. 1c), from which we developed the "distance rules," namely, dsRNA situated 2-nt downstream of a uORF start codon exerts a robust inhibitory effect on mORF translation. Consistent with the distance rule, uORF-ASO_+2, which starts base-pairing at the G of the AUG start codon, was most effective in inhibiting GATA4 protein production ($0.41 \pm 0.04$-fold). Those uORF-ASOs that begin base-pairing either upstream or downstream of this G are less effective in inhibiting GATA4 protein production as a function of their distance from the G residue in AUG (Fig. 3a, middle). The relative inhibitory activities were as follows: uORF-ASO_−3 <uORF-ASO_0 <uORF-ASO_+12 <uORF-ASO_+9 <uORF-ASO_+6 <uORF-ASO_+2 (Fig. 3a, right). These uORF-ASOs that "mimic" the dsRNA element, in this case of *GATA4* mRNA, belong to what we define as Class I uORF-ASOs. Hereafter, the most potent Class I uORF-ASO_+2 is referred to as ASO1 for simplicity (Fig. 3b).

Conversely, we reasoned that uORF-ASOs designed to disrupt the uORF-dsRNA element of cellular *GATA4* mRNA would inhibit uORF translation and, in turn, promote GATA4 mORF protein production. We termed this type of ASOs as Class II uORF-ASOs. Therefore, we designed a *GATA4*-specific Class II uORF-ASO called ASO2 (Fig. 3b). This ASO2, which mimics the MM mutations introduced in the dsRNA as in Fig. 2e, is designed to disrupt the dsRNA structure downstream of the uORF AUG by preventing the two strands of the dsRNA from annealing. We demonstrated the distinct mechanisms of action for two ASOs in influencing mRNA secondary structure using in vitro RNA SHAPE (Supplementary Fig. 3a). The Class I intermolecular dsRNA-forming uORF-ASO, ASO1, diminished the SHAPE reactivity at $G^{384}$-$U^{399}$ downstream of the *GATA4* uORF AUG (Fig. 3b and Supplementary Fig. 3a). In contrast, the Class II dsRNA-disrupting uORF-ASO, ASO2, enhanced the SHAPE reactivity (Fig. 3b and Supplementary Fig. 3a) in the same manner as with the MM mutations (Supplementary Fig. 2c).

To test the specificity and efficacy of these ASOs in regulating the GATA4 uORF activity, we transfected ASO1 and ASO2 in HEK293T cells containing either the WT or the ΔuORF GATA4 5′ UTR-bearing FLuc reporters and the RLuc reference plasmid. Neither ASO produced any effect on the ΔuORF 5′ UTR-bearing reporter (Fig. 3c, d, gray bars). ASO1 specifically suppressed the luciferase activity in the WT 5′ UTR reporter transfected cells (Fig. 3c, pink bars), while ASO2 enhanced the luciferase activity in a dose-dependent manner (Fig. 3d, blue bars).

### uORF-targeting ASOs modulate *GATA4* translation in human cardiomyocytes

We next examined the effect of ASO1 and ASO2 on endogenous *GATA4* mRNA translation in cardiomyocyte cell culture. Transfecting ASO2 in the human immortalized cardiomyocyte (CM) cell line AC16, which expresses GATA4, produced a dose-dependent increase in endogenous GATA4 protein levels (Fig. 3e, left, middle). On the other hand, ASO1 resulted in a dose-dependent decrease in GATA4 protein expression. Neither ASO1 nor ASO2 caused significant changes in *GATA4* mRNA levels following ASO treatment (Fig. 3e, right). ASO1 also did not alter mRNA expression levels of predicted off-targets *LRP1B*, *PCSK5*, *ALG8*, *EHBP1*, *CNKSR2*, and *WWOX* (targeting intronic regions) when allowing for up to a 2-nucleotide mismatch (Supplementary Fig. 3b). More importantly, we found that ASO1 and ASO2 produced no significant global changes in mRNA translation using polysome profiling (Fig. 3f), but rather caused specific shifts of *GATA4* mRNA to lightly translating fractions with ASO1 and to more heavily translating fractions with ASO2 (Fig. 3g and Supplementary Fig. 3c). AC16 cells stained with a β-actin antibody showed shrinkage in cell size with ASO1 treatment and enlargement using ASO2 (Supplementary Fig. 3d), which was consistent with decreased or increased GATA4 protein expression, respectively. Taken together, these results demonstrate that ASOs can be designed to target the mRNA uORF-dsRNA region to control the translation of the *GATA4* mORF in a bidirectional manner.

### uORF-targeting ASOs regulate cellular hypertrophy in human ESC-derived CMs

We further tested these ASOs in CMs derived from human embryonic stem cells (hESCs) (Supplementary Fig. 4a). Human ESC lines with the *GATA4* ΔuORF mutant (heterozygous and homozygous) were generated via CRISPR-Cas9 genome editing with homology-directed repair (Supplementary Fig. 4b). GATA4 protein increased by $1.29 \pm 0.63$-fold in heterozygous mutant hESC-derived CMs and $2.03 \pm 0.17$-fold in homozygous mutant CMs (Supplementary Fig. 4b). We did not observe any comparable difference in *GATA4* mRNA levels. α-actinin and NKX2-5 staining suggest that homozygous ΔuORF ESC-derived CMs exhibited a hypertrophic phenotype when compared to WT CMs (Supplementary Fig. 4c). When treated with the two types of ASOs, WT ESC-derived CMs showed a similar change in GATA4 protein expression as the AC16 CM cell line (Fig. 4a). ASO1 decreased the GATA4 protein level to $0.65 \pm 0.02$ of the control, while ASO2 increased it by $1.93 \pm 0.12$-fold. In contrast, homozygous ΔuORF CMs did not show any changes in GATA4 protein level upon treatment with ASO1 or ASO2 (Fig. 4a), indicating a specific regulatory effect of the two ASOs on *GATA4* uORF activity. Moreover, neither ASO1 nor ASO2 altered *GATA4* mRNA levels, suggesting regulation of translation rather than mRNA stability (Fig. 4b). At the cellular phenotype level, cells transfected with ASO1 exhibited cellular shrinkage, and those treated with ASO2 were hypertrophied, which are in line with respective GATA4 protein level change (Fig. 4c). As an indicator of increased GATA4 protein expression, we found an increase in *MYH6* mRNA (GATA4 target gene) expression in homozygous ΔuORF CMs compared to WT control CMs (Fig. 4d). In addition, we also observed a decrease in *MYH6* mRNA after ASO1 treatment and an increase after ASO2 treatment (Fig. 4e). Moreover, we did not observe any changes in cell number after ASO1 or ASO2 compared to control ASO treatment for 24 h (Supplementary Fig. 4d). These results demonstrate that genetic inactivation of uORF or chemical manipulation of uORF activity using uORF-ASOs regulates GATA4 protein expression and cell hypertrophy in human ESC-derived cardiomyocytes.

### Class I GATA4 uORF-ASO counteracts cardiac hypertrophy in animal models

GATA4 can induce CM hypertrophy in cultured cells in vitro and in adult mouse hearts in vivo via the transcriptional activation of pro-

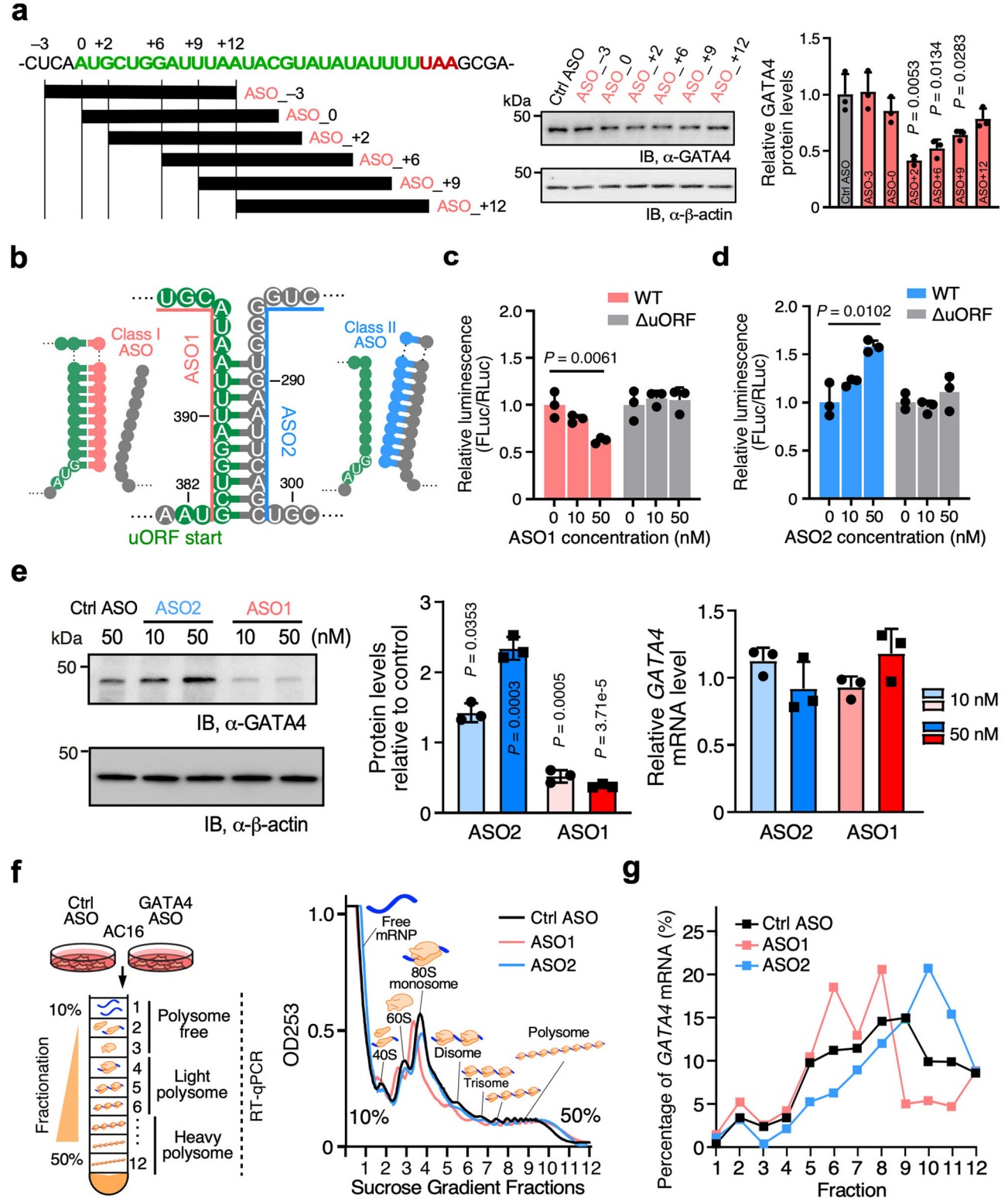

hypertrophic gene expression[24,26,27]. Notably, increased GATA4 protein activity is essential for inducing cardiac hypertrophy during pressure overload in mouse and rat models[29,43,44]. To examine the potential application of uORF-ASOs in vivo, we conducted a proof-of-principle animal study in two cardiac hypertrophy mouse models using the Class I uORF-ASO. In both models, mice were injected via the tail vein with the nanoparticle encapsulated GATA4 ASO1 (Fig. 3b, c), targeting the mouse *Gata4* uORF starting at the 3rd nucleotide of the AUG in vivo. The first animal model is a transverse aortic constriction (TAC) surgery

that induces left ventricle pressure overload and mimics hypertensive cardiomyopathy in humans[45,46]. In this TAC model, WT mice were injected with ASO1 once per week for eight weeks following TAC surgery (Fig. 5a). After eight weeks, the heart weight (HW) to tibia length (TL) ratio, a normalized measure of cardiac enlargement, was significantly reduced from $11.5 \pm 3.5$ mg/mm in control ASO injected TAC mice to $7.8 \pm 1.7$ mg/mm in the ASO1-treated TAC mice (Fig. 5b). The CM cross-sectional surface area was quantified by wheat germ agglutinin staining, marking the boundaries of CMs. Our analysis showed

**Fig. 3 | Mechanism-based design of two types of ASOs for regulating *GATA4* mRNA translation. a** Left: Schematic of screening of several 16-nt 2′-*O*-methylated uORF-targeting ASOs that emulates the presence of dsRNA upstream, at, or downstream of the human *GATA4* uORF start codon. The control ASO used was the mismatch version derived from the ASO_-3. Middle: Western blot analysis of GATA4 protein expression normalized by β-actin in human immortalized AC16 cardiomyocyte cells transfected with 50 nM of ASOs. Right: Quantification of *GATA4* mRNA expression from the middle panel and *ACTB* mRNA is used for normalization. **b** Schematic of two types of ASOs targeting *GATA4* uORF-dsRNA element. Class I (ASO1; the same as ASO_ +2 in **a**) is designed for forming artificial dsRNA downstream of the uORF strand, while Class II (ASO2) is intended to prevent endogenous dsRNA formation and free up the uORF strand. **c, d** Dual luciferase reporter assay with WT and ΔuORF reporters after co-transfection of ASO1 (**c**) and ASO2 (**d**) in HEK293T cells. FLuc activity was normalized to WT and ΔuORF reporter

activity with no ASO1 and ASO2 treatment, respectively. **e** Left and middle: Western blot analysis of GATA4 protein levels normalized to β-actin in human AC16 cells transfected with 10 or 50 nM of ASO1 and ASO2. Total ASO concentrations were equalized to 50 nM using the control ASO. Right: RT-qPCR measurement of relative *GATA4* mRNA level normalized to *ACTB*. **f** Polysome profiles generated from absorbance readings at 253 nm for lysates of AC16 cells transfected with either control ASO, ASO1, or ASO2. **g** RT-qPCR measurement of *GATA4* mRNA distributions across various fractions in polysome profiles. Experiments in (**f**) and (**g**) were repeated three times, and representative data were shown. Data are represented as mean ± SD. *$P < 0.05$, **$P < 0.01$, ***$P < 0.001$; Statistical significance was confirmed by unpaired two-tailed Student $t$ test for **a, c, d**, and 1-way ANOVA followed by Holm-Sidak post hoc test for **e** ($N = 3$ biological replicates). Source data are provided as a Source Data file.

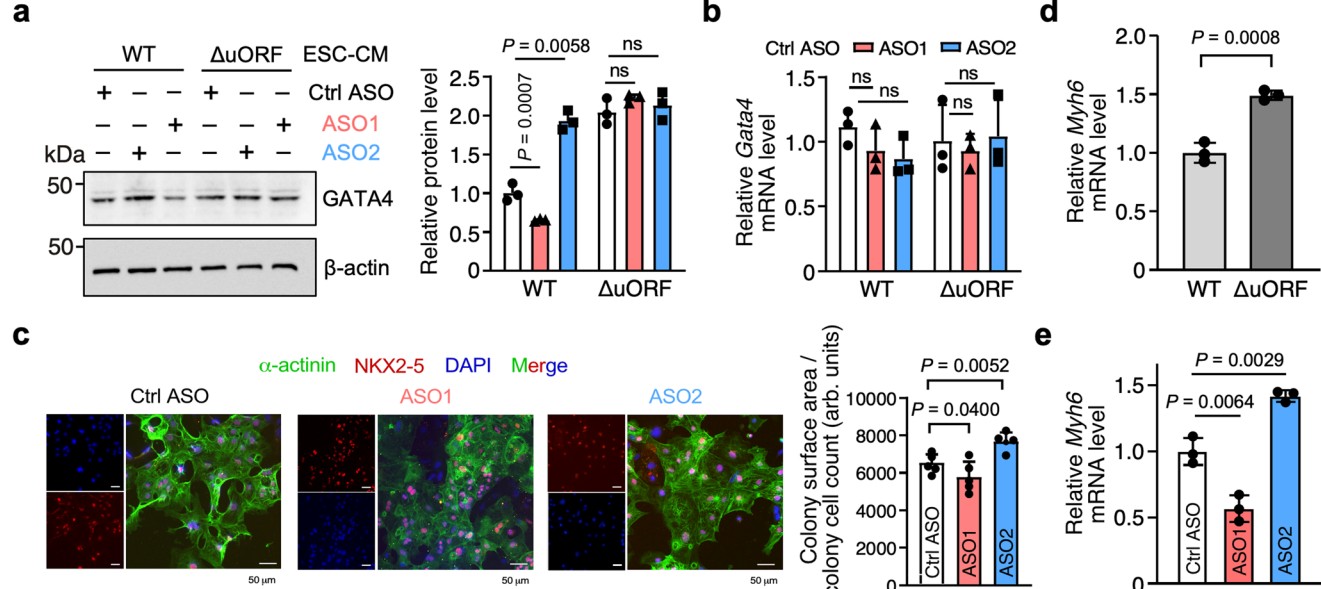

**Fig. 4 | uORF-targeting ASOs regulate GATA4 protein expression and cardiomyocyte hypertrophy in human ESC-derived cardiomyocytes. a** Western blot analysis of GATA4 protein expression in human ESC-derived WT and homozygous ΔuORF CMs with ASO1 and ASO2 (50 nM) or control ASO treatment. **b** RT-qPCR measurement of *GATA4* mRNA normalized to *ACTB*. **c** Representative images of α-Actinin (green) and NKX2-5 (red) immunostaining in addition to DAPI (blue) in ESC-derived CMs treated with control ASO, ASO1, or ASO2. Scale bar: 100 µm. Co-staining of α-actinin (green) and NKX2-5 (red) discerns CMs from mis-differentiated cells. Cell surface area was measured for five different clumps of cells as the total

surface area was divided by the number of cells. **d** *MYH6* mRNA expression in ESC-derived WT and homozygous ΔuORF CMs at baseline. **e** *MYH6* mRNA expression in ESC-derived WT and homozygous ΔuORF CMs with ASO1 and ASO2 or control ASO (50 nM) treatment. Data are represented as mean ± SD. *$P < 0.05$, **$P < 0.01$, ***$P < 0.001$; Statistical significance was confirmed by unpaired two-tailed Student $t$ test for **c** and **d**, and one-way ANOVA followed by Holm–Sidak post hoc test for **a**, **b**, and **e** ($N = 3$ biological replicates). Source data are provided as a Source Data file.

reduced CM hypertrophy in the ASO1 group compared to the control group (Fig. 5c). TAC mice exhibited marked fibrosis as evidenced by the picrosirius red staining, which was reduced in the ASO1 group (Fig. 5d). Cardiac function was also monitored biweekly by echocardiography. Consistent with histological findings, control mice exhibited a lower ejection fraction (32.3 ± 14.8%) compared to the ASO1 injected mice (53.4 ± 17.0%) (Fig. 5e and Supplementary Data 4). The cardiac properties correlated with a ~ 50% reduction in GATA4 protein levels in ASO1 injected versus control mouse heart lysates in the TAC animal studies (Fig. 5f). In contrast, the mRNA level of *Gata4* was not changed (Fig. 5g). Furthermore, CM hypertrophic markers *Nppa* and *Nppb* were significantly reduced in their mRNA levels in the ASO1 group compared to the control group (Fig. 5h).

The second animal model we used here is isoproterenol (ISO)-induced pathological cardiac hypertrophy triggered by neurohumoral β-adrenergic stimulation and increased cardiac workload[45,46]. In this ISO model, wild-type mice were injected with ISO daily for two weeks. Animals also received control ASO or ASO1 once per week for two

weeks, starting concurrently with the initial ISO injection (Supplementary Fig. 5a). After two weeks, the HW to TL ratio was lower in the ISO-treated mice receiving ASO1 injections compared to control ASO-treated mice (Supplementary Fig. 5b). Consistent with the observations in the TAC model, the cellular surface area was significantly reduced in the ASO1 group compared to the control ASO group (Supplementary Fig. 5c, d). In agreement with reduced CM hypertrophy, we observed more than 60% reduction in GATA4 protein levels without any changes in *Gata4* mRNA expression in the ASO1 versus the control ASO group (Supplementary Fig. 5e, f). Again, the GATA4 target gene *Nppa* was significantly reduced in the ASO1-treated mice (Supplementary Fig. 5g). To confirm the delivery of ASO1 in CMs, heart, and potentially other organs, quantitative measurement using a splint ligation-based assay[47] was performed. The data suggested that ASO1 was delivered to the heart and CMs and in multiple other organs, including liver, kidney, and lung, but not to the brain, and it was not retained in the serum (Supplementary Fig. 5h). Since the liver exhibited the highest enrichment of ASO, we tested for signs of organ toxicity by

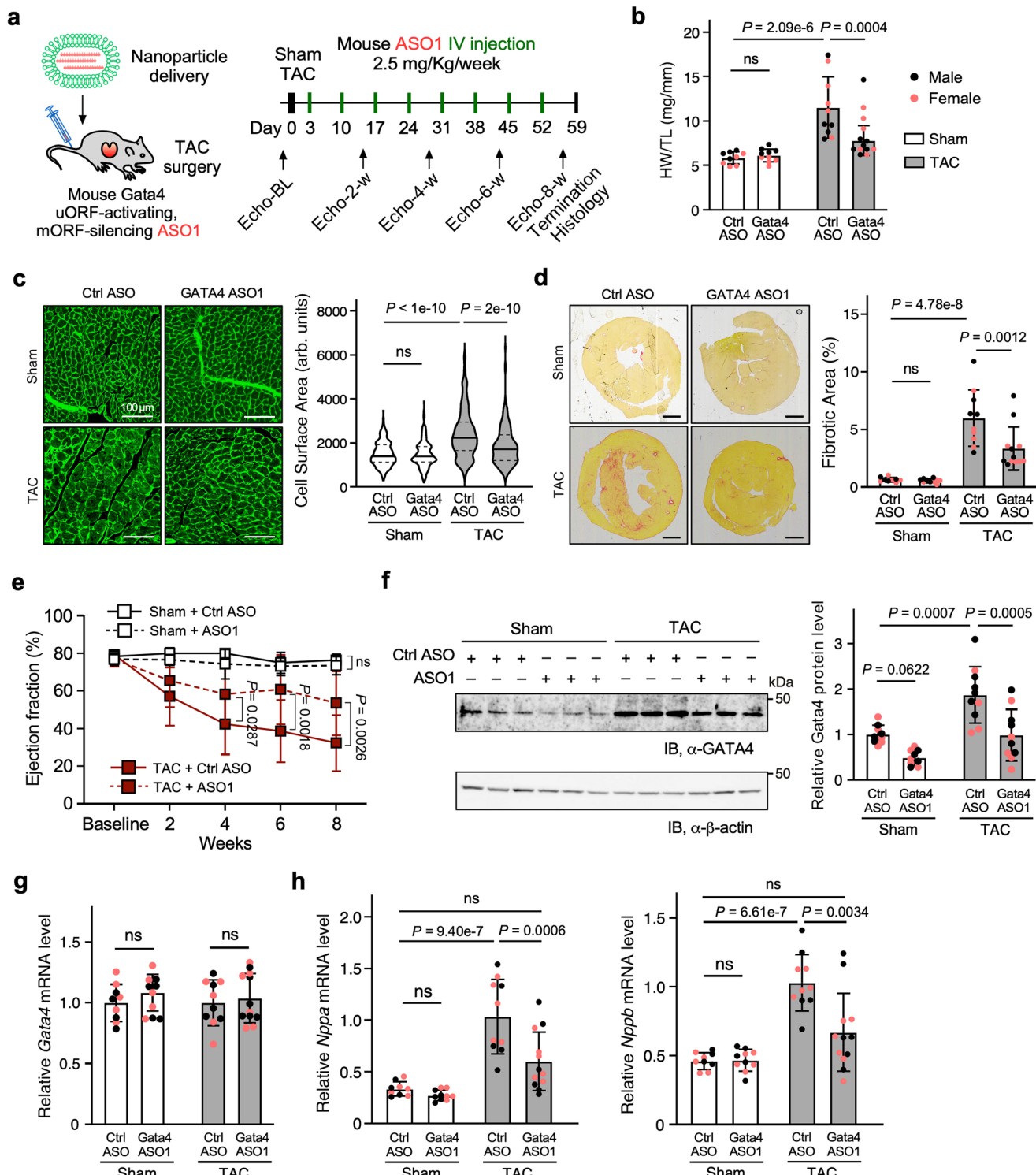

**Fig. 5 | GATA4 uORF-targeting ASOs counteract TAC-induced cardiac hypertrophy. a** Schematic of TAC surgery-induced cardiac hypertrophy mouse model for tail-vein-injected nanoparticle-encapsulated control or uORF-ASOs. Heart samples were then harvested and analyzed. Both sexes were almost equally represented, with male and female dots depicted as black and pink, respectively (**b**–**h**). **b** Cardiac hypertrophy was measured as heart weight normalized to tibia length (HW/TL) in harvested hearts. Ctrl ASO-Sham: *N* = 9; Gata4 ASO-Sham: *N* = 10; Ctrl ASO-TAC: *N* = 10; Gata4 ASO-TAC: *N* = 13. **c** Representative images of wheat germ agglutinin-fluorescein in mouse heart transverse sections highlighting CM perimeter. Scale bars, 20 μm. *N* = 3 hearts with >250 CMs quantified per heart. **d** Representative images of picrosirius red staining of heart sections (left) to quantify the fibrotic area (right). Scale bar: 1 mm. Ctrl ASO-Sham: *N* = 9; Gata4 ASO-Sham: *N* = 10; Ctrl ASO-TAC: *N* = 10; Gata4 ASO-TAC: *N* = 12.

**e** Echocardiographic cardiac function measurements of ejection fraction (EF). Ctrl ASO-Sham: *N* = 9; Gata4 ASO-Sham: *N* = 10; Ctrl ASO-TAC: *N* = 10; Gata4 ASO-TAC: *N* = 13. **f** Western blot analysis of GATA4 protein expression normalized to β-actin in the hearts samples. Ctrl ASO-Sham: *N* = 9; Gata4 ASO-Sham: *N* = 10; Ctrl ASO-TAC: *N* = 10; Gata4 ASO-TAC: *N* = 10. **g** RT-qPCR measurement of *Gata4* mRNA expression normalized to *Actb* mRNA in the hearts. Ctrl ASO-Sham: *N* = 9; Gata4 ASO-Sham: *N* = 10; Ctrl ASO-TAC: *N* = 10; Gata4 ASO-TAC: *N* = 12. **h** RT-qPCR measurement of hypertrophy marker gene mRNA *Nppa* and *Nppb* in mouse heart samples. *Actb* mRNA was used as a normalizer. Ctrl ASO-Sham: *N* = 9; Gata4 ASO-Sham: *N* = 10; Ctrl ASO-TAC: *N* = 10; Gata4 ASO-TAC: *N* = 12. Data are represented as mean ± SD. *P < 0.05, **P < 0.01, ***P < 0.001; Statistical significance was confirmed by two-way ANOVA followed by Holm–Sidak post hoc test for **b**–**h**. Source data are provided as a Source Data file.

measuring the alanine transaminase (ALT) activity in the serum, which was unaffected compared to the control ASO (Supplementary Fig. 5i). Consistent with no obvious systematic toxicity, we did not observe significant changes in mouse behavior and heart rate (Supplementary Data 4) upon treatment of GATA4 ASO1 in comparison with control ASO. These findings demonstrate the in vivo applicability of mechanism-based development of uORF-ASOs to manipulate pathogenic factor expression for potential interventions in cardiac disease treatment.

**Expanding the utility of Class I ASO-mediated intermolecular dsRNA formation to modulate mRNA translation**

Class I and Class II GATA4 uORF-ASOs were evaluated for regulating GATA4 protein expression in AC16 cells and hESC-derived CMs, and Class I uORF-ASOs were tested in wild-type mice in vivo. To enhance the potency of this mechanism-based ASO development, we first sought to modify the length and chemical modifications in the human-specific GATA4 ASO1. GATA4 ASO1 variants with lengths of 16-, 18-, and 20-nts produced similar levels of inhibition of GATA4 protein expression in AC16 cells (Supplementary Fig. 6a). Therefore, we fixed the length of GATA4 ASO1 at 16-nt to examine the effects of including different modified nucleotides in the ASO. The immunoblot results suggest that ASO1 bearing 2′-O-methyl, 2′-O-methoxyethyl (MOE), or phosphorothioate (PS) backbone (Fig. 6a and Supplementary Fig. 6b) produced comparable suppression of GATA4 protein expression (protein levels of $0.42 \pm 0.05$, $0.44 \pm 0.07$, $0.47 \pm 0.01$ times relative to the control, respectively). On the other hand, the locked-nucleic acid (LNA)-based ASO produced a superior reduction of GATA4 protein levels to $0.21 \pm 0.02$ times relative to the control without altering mRNA levels (Fig. 6a, b). Interestingly, the combination of 2′-O-methyl plus four LNA nucleotides at the 3′-end of the ASO exhibited an equally strong mORF-inhibitory effect without affecting mRNA expression levels compared to the LNA-based ASO (Fig. 6c, d).

Because multiple TFs contain uORFs in their mRNAs in humans and mice (Supplementary Fig. 2a), we sought to determine whether we can expand the usage of Class I uORF-ASOs to target mRNAs encoding other proteins, including TFs and non-TFs. We first obtained WT and mutant 5′ UTR-bearing FLuc reporter constructs for multiple cardiac TFs, including human MEF2C (myocyte enhancer factor 2C; two uORFs), NKX2-5 (NK2 homeobox 5; one uORF), and a translation factor, namely, eukaryotic translation initiation factor 4G2 (eIF4G2; one uORF) (see Supplemental Information for specific sequences). Mutations of AUG start codons to UUG in uORFs (ΔuORF mutants) of MEF2C, NKX2-5, and eIF4G2 increased FLuc activity by -1.3–2-fold similarly as for GATA4 without any significant impact on their reporter mRNA expression, suggesting that these uORFs confer mORF suppression (Supplementary Fig. 6c). To determine whether the Class I uORF-ASOs can activate uORF and inhibit mORF translation in these mRNAs besides GATA4, we designed three uORF-ASOs (2′-O-methyl plus four LNA nucleotides at the 3′-end) targeting specific regions in the 5′ UTR of mRNAs of NKX2-5 and eIF4G2 in HEK293T cells together with MEF2C in AC16 cells. Each gene-specific uORF-ASO significantly reduced protein levels of NKX2-5 and eIF4G2 to $0.54 \pm 0.11$ and $0.38 \pm 0.06$ times the control, respectively (Supplementary Fig. 6d). Additionally, the two putative uORFs of MEF2C were tested in a similar fashion. The ASO targeting the first uORF (uORF1) produced a $0.87 \pm 0.11$-fold increase, whereas the ASO targeting the second uORF (uORF2) produced a stronger suppression to $0.53 \pm 0.10$-fold (Supplementary Fig. 6e).

After validating the translational activation effect of Class I uORF-ASOs, we further tested the translation-manipulating effect of Class I ASOs at mORFs in various scenarios to demonstrate the generalizability of these mORF-targeting ASOs (mORF-ASOs). Strikingly, ASOs targeting the cognate mORF AUG initiation codons of mRNAs encoding multiple cardiac TFs, i.e., GATA4, MEF2C, and NKX2-5, produced a

$3.02 \pm 0.15$, $2.61 \pm 0.19$, and $1.54 \pm 0.06$-fold increase in protein levels (Fig. 6e–g). Another unique scenario is eIF4G2 mRNA, whose coding sequence starts with a near-cognate GUG initiation codon[48]. Targeting this eIF4G2 mORF with an mORF-ASO produced a $2.84 \pm 0.24$-fold increase in protein levels (Fig. 6g). Collectively, these findings demonstrate the generalizable applications of Class I ASOs in modulating protein expression at both uORFs and mORFs across multiple mRNAs.

## Discussion

Our initial in vitro studies using artificial reporters reveal that an optimal distance (i.e., 2–5 nt) is required for dsRNA structures to cooperate with the uORF initiation codon and subsequently lead to inhibition of the mORF translation (Fig. 1). A similar reporter system from real-time single-molecule fluorescence spectroscopy showed that a stable hairpin promoted translation initiation of a nearby upstream start codon[49]. Likewise, our reporter assays involving biologically relevant uORF-dsRNA elements of GATA4, MFN1, and DHTKD1 mRNAs agree with this notion. The introduction of mismatch mutations to dsRNA downstream of, and adjacent to, a uORF start codon led to increased FLuc mORF translation (Fig. 2 and Supplementary Fig. 2). A similar effect has been observed in another study whereby mutations were introduced to dsRNA stems of 5′ UTRs[41]. These studies support the idea that dsRNA promotes translation initiation at uORFs. Perhaps this dsRNA regulatory mechanism is an evolutionary measure for controlling translation initiation analogous to a linear Kozak consensus sequence[50] flanking a start codon. Hypothetically, it could be beneficial to tightly regulate dosage-sensitive proteins to prevent normal homeostatic control from going awry. Indeed, this is shown by our results using the GATA4 uORF-ASOs, where a slight perturbation of the dsRNA or uORF in the endogenous GATA4 5′ UTR produced significant changes in GATA4 protein levels, leading to drastic changes in the cellular size of immortalized human CMs (Fig. 3 and Supplementary Fig. 3). These changes in the CM size typically occur only under stress conditions. In addition to the pharmacological ASO-based approach, we also used human ESC-derived CMs with a genetically inactivated uORF (Fig. 4 and Supplementary Fig. 4) to prove the role of uORF in inhibiting the translation of GATA4 mORF and limiting spontaneous cellular hypertrophy.

Our in vitro RRL assays suggest that these effects could be explained by dsRNA sequestration of the PIC, which leads to optimal positioning or an increase in the residence time upstream of the start codons, allowing more efficient PIC recognition of the start codon (Supplementary Fig. 1). This is especially facilitated because the PIC has low helicase activity and may slowly unwind the dsRNA for identifying a start codon. Therefore, RNA helicases may carry out an extra layer of regulation with dsRNA unwinding activities. One example has been shown in yeast, where the expression level of DEAD-box RNA helicase Ded1p (homologous to DDX3X in mammals) controls the extent of 5′ UTR unwinding and results in a reduction of translation initiation events inside 5′ UTR at near cognate start codons[11]. It is still unclear whether a similar mechanism is generally present in human cells. The RNA helicases DDX3X and DHX29 have been shown to play a role in 5′ UTR unwinding and uORF translation in a specific cohort of mRNAs[51,52]. We provided supporting evidence that DDX3X may modulate uORF-dsRNA activity and function at baseline and under specific stress conditions such as LPS-induced inflammatory stress (Supplementary Figs. S2i–n).

This work offers a rational approach for developing translation-manipulating ASOs. Our understanding of uORF-dsRNA synergy rules allows the design of two classes of uORF-ASOs (Figs. 1, 3). The Class I uORF-ASOs that form a more stable artificial intermolecular dsRNA with a region immediately downstream of a uORF of a target mRNA recapitulate our observed uORF-dsRNA relationships and suppress the mORF translation (Fig. 3 and Supplementary Fig. 3a, 6f). The PIC may

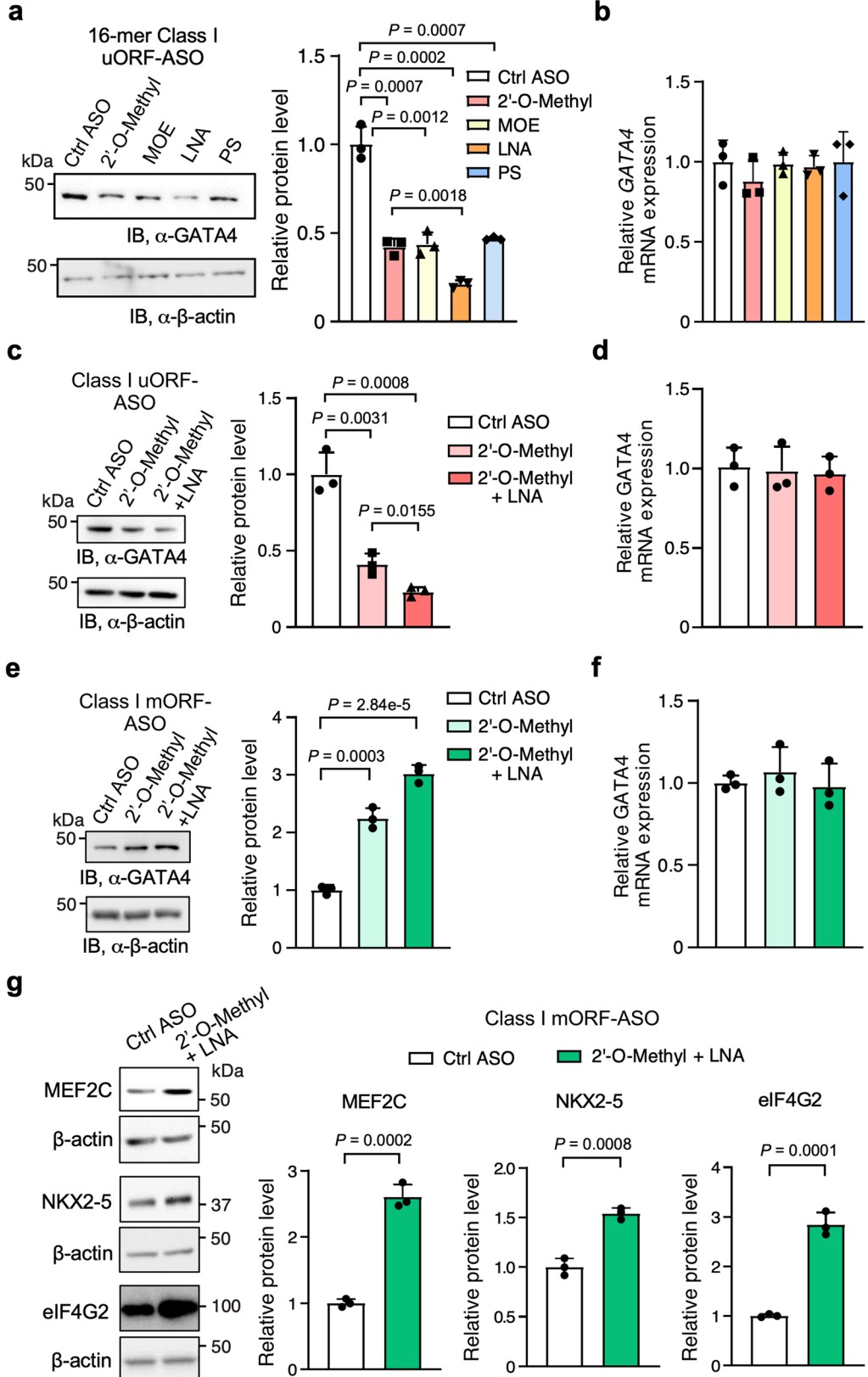

pause upstream of the dsRNA with extended dwelling time and allow more efficient recognition of uORF start codon and joining of 60S ribosome large subunit, therefore promoting the translation of uORF and inhibiting mORF translation. In this dynamic process, as the ASO-mRNA duplex forms immediately downstream of the uORF AUG start codon in *GATA4* mRNA 5′ UTR, the PIC likely unwinds the ASO-mRNA

intermolecular duplex, possibly using an intrinsic component factor, namely, the RNA helicase eIF4A within the PIC[53], together with DDX3X[11,38] (Supplementary Fig. 2i−n). This allows the 80S ribosome to start the translation elongation on the uORF sequence. Presumably, this potential helicase unwinding process during PIC scanning should be acutely sensitive to melting temperature (Tm), the nature of the 2′

**Fig. 6 | Enhancing and expanding the functionality of Class I ASOs to modulate the translation of various mRNAs. a–d** Enhancing the uORF-targeting ASO1 through various nucleotide chemistries. Locked nucleic acid (LNA) bases produced greater suppression of GATA4 protein levels compared to 2′-*O*-methylated, 2′-*O*-methoxy-ethyled (MOE), and phosphorothioate (PS) backbone (**a**). The combination of 2′-*O*-methyl and LNA (4 LNA at ASO 3′ end) is superior to 2′-*O*-methyl alone (**c**). RT-qPCR measurement of *GATA4* mRNA in (**a**) or (**c**) with *ACTB* mRNA used as a normalizer (**b**, **d**). **e–g** Examining effects of class I mORF-enhancing ASOs (mORF-ASOs) on mORF translation. GATA4 mORF-targeting ASOs enhance its mORF

protein levels (**e**). Like with uORF-specific ASOs, the combination of 2′-*O*-methyl and LNA is superior to 2′-*O*-methyl alone and does not change mRNA levels (**f**). 2′-*O*-methyl and LNA (4 LNA at ASO 3′ end) mORF ASOs (50 nM) increase the protein levels of mRNAs with cognate start codons, in the case of *MEF2C* (in AC16 cells) and *NKX2-5*, or near-cognate GUG start codons as with *EIF4G2* (in HEK293T cells) (**g**). Data are represented as mean ± SD. *$P < 0.05$, **$P < 0.01$, ***$P < 0.001$; Statistical significance was confirmed by unpaired two-tailed Student $t$ test for **a–g** ($N = 3$ biological replicates). Source data are provided as a Source Data file.

modifications used, and the placement of the modifications[41,54]. This is supported by our finding that 2′-*O*-methyl plus four LNA nucleotides at the 3′-end of the ASO and LNA-based ASO show stronger translation repressive activity compared to 2′-*O*-methyl-, MOE-, or PS-based ASO for GATA4 ASO1 (Fig. 6a–d). Furthermore, chemical modifications and their positions may be sensitive to target mRNA and ASO sequences, the nature of the mRNA-ASO duplexes formed, the length of the ASO, the strength of the uORF and mORF (e.g., Kozak sequence context), and nearby structures such as dsRNA or other translation inhibitory elements[41,54].

In contrast, the Class II uORF-ASOs that disrupt the endogenous dsRNA downstream of, and adjacent to, a uORF, recapitulate our dsRNA mismatch mutations or DDX3X-mediated dsRNA unwinding and alleviate uORF suppression of the mORF (Fig. 3 and Supplementary Figs. 3a and 6f). In this case, the PIC may skip the uORF start codon and reach the mORF start codon to assemble into an 80S ribosome to translate the primary CDS and synthesize the main protein product. The Class II ASOs require the presence of a dsRNA downstream of the uORF start codon (Fig. 3 and Supplementary Fig. 6f). We characterized the dsRNA within the *GATA4* 5′ UTR and showed its disruption via ASOs could increase GATA4 protein levels (Figs. 3 and 4). Other studies have demonstrated similar effects by disrupting dsRNA to boost protein expression[16,41]. However, it is unclear how exactly these targeted dsRNAs are inhibitory in these cases. Our work has provided insights into the mechanism of action for uORF-dsRNA-mediated translational control and the application of our "distance rule" to develop mechanism-based ASOs. Class I and Class II ASOs can be utilized to control protein levels for various applications, and more studies are needed to elucidate the dichotomy between these two targeting mechanisms.

The Class I uORF-ASOs have proven to be versatile in their uses because they form artificial translation-enhancing dsRNA with the linear target mRNA sequence without relying on the presence of existing secondary structures. To our surprise, this can be extended to mORFs to activate their translation (Supplementary Fig. 6f), offering a unique opportunity to increase protein levels. Strikingly, ASOs directly targeting the mORF of *GATA4*, *MEF2C*, *NKX2-5* (which possess moderately strong Kozak sequences) and *eIF4G2* (which bears a GUG near-cognate start codon) lead to appreciable increases in their protein expression (Fig. 6e–g). The fact that these ASOs target the mORF suggests that mORF start codons are inherently "leaky" (i.e., could be skipped by the PIC), and that ASOs could aid in more efficient translation initiation. Previous studies showed that antisense cDNA annealing with the mORF downstream of the start codon improves its translation using an RRL in vitro translation system[55]. A recent study reported that the ribosome could translate internal ORFs (iORF) within CDS[56], suggesting that some leakiness in certain mORF start codons is permissible and that initiation at mORF start codons is occasionally suboptimal and might be artificially enhanced.

These mechanism-based ASOs could decrease harmful proteins or increase beneficial ones. In our case, administration of Class I uORF-ASOs targeting the *GATA4* uORF in human cells and mice reduced GATA4 protein levels. Human CMs experienced an atrophy phenotype (Fig. 4c and Supplementary Fig. 3d). Whereas, in mice that have undergone TAC surgery or ISO injection (Fig. 5 and Supplementary

Fig. 5), this reduction coincided with resistance to cardiac hypertrophy, recapitulating phenotypes in *Gata4* conditional knockout mice[30]. Intriguingly, we observed improved cardiac function after this short-term ASO treatment in our TAC model. It is still unclear if this could reverse existing heart disease and offer a viable therapeutic approach. Our work shows that these ASOs can be used in other therapeutic applications. For instance, our Class I mORF-ASOs can increase protein levels in a manner that is simpler than viral delivery methods, especially for large genes (e.g., Titin) (Supplementary Fig. 6f). Long-standing needs for overexpressing therapeutic proteins exist to treat diseases caused by genetic haploinsufficiency or aberrant expression[57]. Alternatively, cell identity switch is a promising approach to improve organ function and reverse disease progressions, such as cardiac fibroblast-to-CM trans-differentiation driven by overexpressing a cocktail of TFs, including GATA4, MEF2C, TBX5, and NKX2-5[58–60]. We provided "proof-of-concept" evidence to support the idea of increasing protein levels of GATA4, MEF2C, and NKX2-5 using the Class I mORF-ASOs (Fig. 6g), which may serve as a "booster" to enhance the efficacy of the TF cocktail.

In summary, our work lays the foundation for understanding how dsRNA within a 5′ UTR, including within an uORF, can work in union with the uORF start codon to suppress mORF translation. It highlights various applications of targeting this mechanism by ASOs, which can serve as biochemical tools to study RNA regulatory elements or control protein expression. Through the mechanism-based development of translation-manipulating ASOs, it is possible to produce potential therapeutics for disease treatment.

The dsRNA elements we tested are potentially bound by dsRNA-binding proteins or unwound by RNA helicases to regulate structural stability. This could add an extra layer of regulation not accounted for in this study. Targeting RNA using ASOs may interfere with these RNA-protein interactions, which can contribute positively or negatively to regulating the dsRNA structure and mRNA translation. Notably, GATA4 is an essential homeostatic cardiac protein for CM survival under chronic stress conditions. It is unknown whether reducing GATA4 using ASOs would lead to long-term repercussions on cardiac health[29,31]. More work is needed to study the long-term effect of these ASOs in cells or organs to elucidate the therapeutic potential in our proof-of-principle study, which was primarily intended to test the use of ASOs as gene expression manipulation tools.

## Methods

### Materials and reagents

Information on chemicals, reagents, kits, plasmids, and cell lines used in this study is provided in Supplementary Data 7.

### Molecular cloning

WT and mutant firefly luciferase reporters were made based on the backbone plasmid purchased from Addgene (https://www.addgene.org/114670/). PCR amplified the 5′ UTR DNA fragments from cDNA prepared from HEK293T or AC16 human CM cell line using primers with the extra 5′ end corresponding to the BsmBI cut sites in the plasmid: "AACGTCTCCACAC" for the forward primer and "AACGTCTCTCTTCCAT" for the reverse primer. The DNA fragments were cleaved with the backbone via the restriction enzyme BsmBI for

1 h at 55 °C and then ligated using T4 DNA ligase at room temperature, followed by selection on X-Gal coated plates. Colonies that were not blue were cultured and verified using sequencing. When needed, the plasmids were further mutagenized using PCR. Q5 high-fidelity polymerase was used to introduce the desired mutations. 0.5 μl of the PCR reaction was then incubated with a mixture of T4 DNA ligase, T4 PNK, and DpnI in T4 DNA ligase buffer for 1 h at 37 °C. 5 μl of this reaction was transformed into competent cells and then plated on ampicillin agarose plates. For C-terminal or N-terminal 3x FLAG-tagged GATA4 uORF cloning, primers were used to add 3x FLAG tags to the uORF using PCR following by regular cloning protocol as above. The primer information is included in the Supplementary Data 5. mRNA sequence information for target genes is included in Supplementary Data 8.

### Animal work
C57BL/6J mice of the same age (10–12 weeks) and gender (male and female) from littermates or sibling mating were used for experiments with WT mice. All animal procedures were performed in accordance with the National Institutes of Health (NIH) and the University of Rochester Institutional guidelines. The University of Rochester Medical Center Animal Care and Use of Committee approved all experimental animal procedures. This study used wild-type (WT) C57BL/6J mice (Jackson Laboratories). Mice were maintained on a 12-hour light/dark cycle and fed with a normal chow diet and water at 22 °C with 40–60% humidity in a vivarium facility. This study used two mouse heart failure models: isoproterenol (ISO) injection and transverse aortic constriction (TAC) surgery. We used age-matched male and female mice in the study at ~8–12 weeks. All the mouse surgeries were done by the mouse Microsurgical Core facility at URMC. The mice were randomized for experiments using simple randomization with a specific ID number before animal procedures. All animal operations, including isoproterenol (ISO) injection, transverse aortic constriction (TAC) surgery, and echocardiography measurement, were performed by the Microsurgical Core surgeons following previous protocols[45,46] and full methods in Supplemental Information. The Histology Core did sections and histology analysis. The technicians from both Microsurgical Core and Histology Core were all blinded to the genotypes of the mice and tissue samples.

### In vivo therapeutic model
Based on recommendations from the nanoparticle user instruction, antisense oligonucleotides (ASOs) with chemical modification (resistant to nuclease degradation in vivo) were used per injection. In our experiments, mismatched control and GATA4-specific ASOs were used at the dose of 2.5 mg/Kg body weight in the volume of 150–200 μl for injections in 8–12 weeks old male and female WT C57BL/6 J mice (ISO or TAC model). The ASOs were dissolved in ~150–200 μl RNase-free water. The diluted ASOs were incubated with 50 μl of nanoparticle-based in vivo transfection reagent in sterile tubes for 20 min at room temperature. Transfection enhancer (10 μl) was added to the mixture, vortexed gently, and incubated for 5 min at RT. The nanoparticle-ASO complex was mixed with an appropriate volume of the sterile solution of 5% glucose (w/v), and delivered by intravenous tail vein injections after mice were anesthetized using 2.0% isoflurane[46]. In our heart disease models, ASO injections were started at the time of initial ISO injection or 3 days post-TAC surgery. Injections were carried out once a week for two weeks in the ISO model and eight weeks for the TAC model. Mice were injected with nanoparticles carrying ASOs (2.5 mg/Kg body weight) through the tail vein once a week using a 1 mm BD insulin syringe.

### Echocardiography
For the TAC surgery mouse model, M-mode short-axis echocardiographic image collection was performed using a Vevo2100 echocardiography machine (VisualSonics, Toronto, Canada) and a linear-

array 40 MHz transducer (MS-550D). Heart rate was monitored during echocardiography measurement. Image capture was performed in mice under general isoflurane anesthesia with heart rate maintained at around 550–650 beats/min. The HR could vary in individual mice due to the potential effect of anesthesia or the surgeon's operation variation. LV systolic and diastolic measurements were captured from the parasternal short axis in M-mode. Fraction shortening (FS) was assessed as follows: % FS = (end diastolic diameter - end-systolic diameter) / (end diastolic diameter) x 100%. Left ventricular ejection fraction (EF) was measured and averaged in both the parasternal short axis (M-Mode) using the tracing of the end-diastolic dimension (EDD) and end systolic dimension (ESD) in the parasternal long axis: % EF = (EDD-ESD)/EDD. Hearts were harvested at multiple endpoints depending on the study. In addition to EF and FS, left ventricular end-diastolic diameter (LVEDD), left ventricular end-systolic diameter (LVESD), and wall thickness of left ventricular anterior (LVAWT) and posterior (LVPWT) were also assessed.

### Cell culture and transfection
Human HEK293T cells were propagated in Dulbecco's modified Eagle's medium (DMEM), and AC16 cells were propagated in an equal mix of F12 and DMEM media. Both media were supplemented with 10% fetal bovine serum (FBS), 2 mM L-Glutamine, and 1x Penicillin/Streptomycin Solution. AC16 cells were purchased from Sigma and authenticated, and all the cells were tested for mycoplasma contamination using a detection kit. For expression of N- or C-terminal FLAG-tagged GATA4 uORF peptide, the plasmid was transfected into 6-well plates using Lipofectamine 3000. Cells were harvested 24 h after transfection.

When transfecting ASOs for Western blot, $5 \times 10^5$ AC16 or HEK293T cells were seeded in 10 cm dishes. Once adhered overnight, the culture medium was changed to OPTI-MEM and transfected with 10 nM or 50 nM ASOs using RNAiMAX following the manufacturer's guidelines for 6 h, after which the medium was changed back to the regular culture medium. Cells were harvested 24 h after transfection.

For dual-luciferase assays, HEK293T cells were seeded in 96-well plates at a density of $1 \times 10^4$ cells per well and left to adhere overnight. The cells were then transfected each with a 5' UTR-FLuc reporter plasmid (50 ng) and a control RLuc plasmid (10 ng) using Lipofectamine 3000 for 24 h based on manufacturer's guidelines.

The H7 human embryonic stem cells (ESC) were cultured on Matrigel matrix-coated 6-well cell culture plates. The cells were maintained in a StemFlex medium at 37 ˚C with 5% oxygen and 7.5% $CO_2$ with a complete medium change every other day. Every four days, the colonies were dissociated with enzyme-free passaging reagent ReLeSR. The cell aggregates were then seeded on Matrigel matrix-coated surface at the desired density. The transfection of ASO into ESC-differentiating cardiomyocytes was achieved using Lipofectamine RNAiMax. For each transfection in 6 wells, 7.5 μL of Lipofectamine RNAiMax and ASOs were diluted in 150 μL OPTI-MEM, achieving a 50 nM concentration in culture. Then, the diluted Lipofectamine and ASOs were mixed and incubated at room temperature for 5 min. After incubation, the mixture was added to the cell culture. The transfection was performed three times during the cardiomyocyte differentiation, on Day 1, Day 3, and Day 5 after the medium replacements.

When transfecting siRNA for gene knockdown, siRNAs (50 nM) against DDX3X, eIF1, eIF5, DENR, and DHX29 were transfected in HEK293T cells using Lipofectamine 3000 following the manufacturer's instructions.

### Cell proliferation measurement
For cell proliferation, cells were seeded at 500/well in 96-well plates as biological triplicates. MTT assay was performed using MTT Cell Proliferation Kit I per the manufacturer's recommendations at 0 h and at 24 h.

## Co-immunoprecipitation

Cells from one ~80% confluent 10-cm culture dish were harvested using a native lysis buffer (300 µL) with 50 mM HEPES (pH 7.5), 150 mM KCl, 2 mM EDTA, 0.1% (v/v) NP-40, 1 mM DTT, and 25 µl/ml protease inhibitor cocktail and RNasin RNase inhibitor. Immunoprecipitation (IP) of endogenous DDX3X was performed using 1 µg (dilution factor 1:125) rabbit anti-human DDX3X antibody followed by magnetic dynabeads protein G pull-down. Beads were mixed with 70 µl of SDS loading buffer and boiled for 5 min to elute proteins for western blot. The antibody information is included in the Supplementary Data 6.

## RNA-binding protein immunoprecipitation

Cells from one ~80% confluent 10-cm culture dish were harvested in native lysis buffer (300 µL) with 50 mM HEPES (pH 7.5), 150 mM KCl, 2 mM EDTA, 1 mM NaF, 0.1% (v/v) NP-40, 1 mM DTT, and RNasin RNase inhibitor. Immunoprecipitation of endogenous DDX3X was performed using 1 µg (dilution factor 1:125) rabbit anti-human DDX3X antibody followed by Magnetic protein dynabeads G pull-down. Bound RNA was extracted from IP beads following the Trizol extraction protocol for RT-qPCR. The primer information is included in the Supplementary Data 5.

## Adult cardiomyocyte isolation

Adult cardiomyocytes (CMs) were isolated from 2–4 months old male and female mice using a Langendorff perfusion system as previously described[45]. Mice were anesthetized via intraperitoneal injection of ketamine/xylazine. The heart was excised and fastened onto the CM perfusion apparatus and perfusion was initiated in the Langendorff mode. Our Langendorff perfusion and digestion consisted of 3 steps at 37 °C: 4 min with perfusion buffer (0.6 mM $KH_2PO_4$, 0.6 mM $Na_2HPO_4$, 10 mM HEPES, 14.7 mM KCl, 1.2 mM $MgSO_4$, 120.3 mM NaCl, 4.6 mM $NaHCO_3$, 30 mM taurine, 5.5 mM glucose, and 10 mM 2,3-butanedione monoxime), then switched to digestion buffer (300 U/ml collagenase II [Worthington] in perfusion buffer) for 3 min, and finally perfused with digestion buffer supplemented with 40 µM $CaCl_2$ for 8 min. After perfusion, the ventricle was placed in a sterile 35 mm dish with 2.5 ml digestion buffer and shredded into several pieces with forceps. 5 ml stopping buffer (10% FBS, 12.5 µM $CaCl_2$ in perfusion buffer) was added and pipetted several times until tissues dispersed readily, and the solution turned cloudy. The cell solution was passed through a 100 µm strainer. CMs were settled by incubating the cell suspension at 37 °C for 30 min. The CMs were resuspended in 10 ml stopping buffer and subjected to several steps of calcium ramping: 100 µM $CaCl_2$, 2 min; 500 µM $CaCl_2$, 4 min; 1.4 mM $CaCl_2$, 7 min. Then the CMs were seeded onto a glass bottom dish (Nest Biotechnology) pre-coated with laminin (ThermoFisher Scientific). Plates were centrifuged for 5 min at 1,000 g at 4 °C to increase the adherence, cultured at 37 °C for ~1 h, and then switched to adult CM culture medium (MEM [Corning] with 0.2% BSA, 10 mM HEPES, 4 mM $NaHCO_3$, 10 mM creatine monohydrate, 1% penicillin/streptomycin, 0.5% insulin-selenium-transferrin, and 10 µM blebbistatin for cell culture.

## ASO quantification

Mice were injected with either the nanoparticle encapsulated control ASO or Gata4 ASO via the tail vein. After 48 h, the mice were anesthetized and 100 µL of blood was drawn and left to coagulate. The coagulated material was then spun at 15,000 RCF for 10 min to obtain the serum as the supernatant. The heart, lung, liver, spleen, brain, and kidney were also excised. Primary cardiomyocytes were isolated from the mouse heart as described above[46]. RNA of the isolated organs or cells was extracted using Trizol. The ASO was quantified using an established method using a splint ligation-based assay[47]. In brief, the isolated RNA was incubated with flap oligo A, which contains 3′ sequence that base pairs with the 3′ 9 nucleotides of the Gata4 ASO,

and flap oligo B, which base pairs with the remaining 7 nucleotides of the ASO. The annealed product was incubated with SplintR ligase which ligates both flap oligos together only when they are base pairing with the ASO. The final ligated product reaction was then used as the template in qPCR reaction with the probe that base pairs with the ligated region. The information on ASOs and probes for ASO quantification is included in the Supplementary Data 5.

## Western blotting

Cells were lysed in RIPA buffer. Total cell proteins were separated in a 6–15% denaturing polyacrylamide gel and transferred to polyvinylidene difluoride membranes (PVDF), and probed using the designated primary antibodies (e.g., for GATA4 and β-actin). Next, the membranes were incubated with the appropriate mouse or rabbit secondary antibody and were used to conjugate with horseradish peroxidase (GE Biosciences). Blots were quantified using Image J (NIH). To detect N- and C-terminal FLAG-tagged GATA4 uORF peptides, HEK293T cell lysates samples were prepared for traditional western blotting following a previous protocol[61]. The uncropped and unprocessed scans of the western blots were provided in the Source Data file. The antibody information is included in the Supplementary Data 6.

## Dual-luciferase reporter assay

Transfected cells were incubated with Dual-Glo luciferase substrate according to the manufacturer's recommendations. The final readings of the FLuc were normalized to RLuc to obtain the relative luminescence reading. Folding free energy changes for hairpin stem-loops were estimated using the efn2 program in RNAstructure with default settings. The folding free energy changes include the terminal mismatches at the base of the stem, which are known to stabilize folding.

## In vitro ribosome loading and sucrose gradient fractionation in RRL

To explore the mechanism underlying the dsRNA-mediated influence of uORF translation on mORF translation, we internally labeled four in vitro-transcribed mRNAs using the [α-$^{32}$P]-ATP (250 mCi) and added poly(A) tails for the sequences below using HiScribe™ T7 ARCA mRNA Kit (with tailing). The RNA was extracted as described in RNA SHAPE. Each RNA (5 µl) was added to rabbit reticulocyte lysate reactions according to the manufacturer's recommendations and incubated for 30 min at 30 °C. The whole lysate was added to 15% sucrose gradients and centrifuged at 150,000 g for 2 h and 20 min. The gradients were fractionated by hand into 12 tubes. The radioactivity was quantified using liquid scintillation for each tube. The profile for an individual mRNA indicates the fraction of the sum of the total radioactivity across each tube.

## RNA purification and RT-qPCR

Media was aspirated from adherent cells and washed twice with chilled PBS. The cells were lysed by adding 1000 µl of Trizol directly to the cells, which were mixed with 200 µl of chloroform and incubated for 5 min on ice. The mixture was centrifuged at 16,000 g for 10 min. RNA was precipitated from the aqueous layer by adding two volumes of isopropanol and centrifuged at 16,000 g for 10 min. The pellet was washed twice with 70% ethanol, left to dry, and resuspended in nuclease-free water. To quantify mRNA levels, cDNAs were prepared using iScript master mix RT Kit and qPCR-amplified using SYBR Primer Assay kits. Notably, when a primer set was first used, the identity of the resulting PCR product was confirmed by cloning and sequencing. The quantitative nature of each primer was also assessed by performing a standard curve of varying cDNA amounts. Once confirmed, melting curves were used in each subsequent PCR to verify that each primer set reproducibly and specifically generates the same PCR product. The primer information is included in the Supplementary Data 5.

## Polysome profiling and RNA extraction

Cells were incubated with cycloheximide (100 µg/ml) for 10 min and harvested using a native lysis buffer with 100 mM KCl, 5 mM MgCl₂, 10 mM HEPES, pH 7.0, 0.5% IGEPAL® CA-630, 1 mM DTT, 1 U/µl RNasin ribonuclease inhibitor, 2 mM vanadyl ribonucleoside complexes solution, 20 µl protease inhibitor cocktail (50x), cycloheximide (100 µg/ml). The lysate was centrifuged at 1500 g for 5 min to pellet the nuclei. The supernatant was loaded onto a 10–50% sucrose gradient and centrifuged at 150,000×*g* for 2 h and 20 min. The gradients were transferred to a fractionator coupled to an ultraviolet absorbance detector that outputs an electronic trace across the gradient. Using a 60% sucrose chase solution, the gradient was pumped into the fractionator and divided equally into 12 fractions. RNA was extracted by mixing 500 µl of each fraction with equal volumes of chloroform: phenol: chloroform: isoamyl alcohol (25:24:1) and 0.1 volumes of 3 M sodium acetate (pH 5.2), then centrifuged at 16,000 g for 10 min. The upper aqueous layer was used to repeat this extraction process. The final upper aqueous was then mixed with two volumes of 100% ethanol and left to incubate at −20 °C overnight. The solution was centrifuged at maximum speed for 30 min to pellet the RNA, which was washed twice with 70% ethanol, and finally resuspended in nuclease-free water.

## Selective 2′ hydroxyl acylation analyzed by primer extension assay

The RNA was transcribed using HiScribe T7 Quick High Yield RNA Synthesis Kit. The RNA was purified as discussed above and reconstituted in RNase-free water. Purified RNA from in vitro transcription was heated in RNase-free water for 2 min at 95 °C, then flash-cooled on ice. A 3× SHAPE buffer [0.333 M HEPES (pH 8.0), 0.02 M MgCl₂, 0.333 M NaCl] was added, and the RNA was equilibrated at 37 °C for 10 min with or without ASOs. The RNA was then incubated at 37 °C for 15 min. To this mixture, 1 µL of 10 × NAI (2-methylnicotinic acid imidazolide) stock in DMSO (+), or DMSO alone (−), was added to a final concentration of 25 mM. The NAI reaction proceeded for 15 min. RNA was extracted and reverse transcribed using two CY5-labeled primer sequences (in the reagent table list), and visualized using 8% UREA (8 M) PAGE[33]. The gel bands were quantified in SAFA (Semi-Automated Footprinting Analysis)[62]. The SHAPE reactivity values were cleared of outliers and normalized. Then, the final values were used as SHAPE constraints for the RNA Fold web server below[63]. The code for SHAPE normalization is included in Supplementary Data 9.

## Immunofluorescence and confocal microscopy

Immunostaining of cells grown on coverglass or chambered slides: AC16 cells and ESC-derived CMs were grown on the coverslips for 24 h at 37 °C before being fixed for 10 min with 4% paraformaldehyde in PBS. Cells were washed with PBS for 3 × 5 min and permeabilized using ice-cold 0.5% Triton X-100 in PBS for 5 min. After blocking with 1% BSA in PBS, the coverslips were incubated with indicated primary antibodies (anti-α-actinin: 1:1000; anti-NKX2-5: 1:500; anti-β-actin: 1:2000) in the blocking solution (2% BSA in PBS) for 1 h at RT and then washed with PBS for 3 × 5 min. The coverslips were incubated with the Alex Fluor-488 conjugated secondary antibodies (1:1000) in PBS and washed with PBS for 3 × 5 min. Coverslips were air-dried and placed on slides with an antifade mounting medium (containing DAPI). The slides were imaged using an Olympus FV1000 confocal microscope.

## Wheat germ agglutinin staining

The wheat germ agglutinin (WGA, 5 mg) was dissolved in 5 ml PBS (pH 7.4). We performed deparaffinization by following steps: (i) Xylene (100%) for 2 × 5 min; (ii) Ethanol (100%) for 2 × 5 min; (iii) Ethanol (95%) for 5 min; (iv) ddH₂O for 2 × 5 min. The slides were kept in a pressure cooker for 10 min, along with citrate buffer (10 mM, pH 6.0) for antigen retrieval. We quenched the slides with 0.1 M glycine in phosphate buffer (pH 7.4) for 1 h at RT. Circles were made with a Dako pen, and slides were blocked with normal goat serum for 30 min. In all, 10 µg/ml of WGA-Alexa Fluor 488 was applied to the slides for incubation for 1 h at RT. Slides were rinsed in PBS 3 × 5 min. A coverslip was placed on the slides with VECTASHIELD HardSet antifade mounting medium with DAPI for imaging. Five different cross-sectional areas were selected, and the cell size of at least 500 CM cells was measured per area.

## Statistics

All quantitative data were presented as mean ± SD and analyzed using GraphPad Prism 8.3.0 software (GraphPad). For a comparison between two groups, an unpaired two-tailed Student t-test for normally distributed data was performed. For a comparison between more than two groups, ANOVA followed by the Holm-Sidak multiple comparison test was used to determine the statistical significance among groups. Two-sided P values < 0.05 were considered to indicate statistical significance. Specific statistical methods and post hoc tests are described in the relevant figure legends.

## Reporting summary

Further information on research design is available in the Nature Portfolio Reporting Summary linked to this article.

## Data availability

Further information and stable reagents generated in this study are available from the corresponding author upon request. All raw gel and blot images and graph data for main and supplementary figures are provided in the Source Data file. All nucleotide sequences used in the manuscript are available in Supplementary Data 5 and 8 (NCBI Nucleotide database accession numbers are included). Sequences of the human mRNA 5′ UTRs in Fig. 1a, b were retrieved from GRCh38.p13 (https://ftp.ebi.ac.uk/pub/databases/gencode/Gencode_human/release_40/gencode.v40.transcripts.fa.gz). Source data are provided with this paper.

## Code availability

Code for SHAPE normalization is provided in Supplementary Data 9. RNAstructure is open source and freely available under the GPL V2 license at http://rna.urmc.rochester.edu/RNAstructure.html.

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

## Acknowledgements

We appreciate Emily Bonanno for critically reading and polishing the manuscript and appreciate the technical assistance from Erika Flores Medina in histology and Deanne Mickelsen in surgical operations (Aab CVRI). We appreciate the consulting contributions from Lynne Maquat, Eric Small, Mitchell O'Connell, and Juilee Thakar. We thank Jared Hollinger for general management and technical support. This work was supported in part by the National Institutes of Health (R01 HL132899, R01 HL147954, R01 HL164584, and R01 HL169432 to P.Y.), NIH T32 Fellowship (T32 GM068411 to O.H.), R01 GM132185 and R35 GM145283 to D.H.M., R21NS104878, Link Foundation to L.X. and C.P., the Career Development Award [848985 to J.W.] from the American Heart Association, Harold S. Geneen Charitable Trust Award from the Medical Foundation at Health Resources in Action, LeapRx Program Award 2022 from Empire Discovery Institute and Novo Nordisk, University Research Award from University of Rochester, and start-up funds and Rubens Discovery Award 2021 from Aab Cardiovascular Research Institute of University of Rochester Medical Center (to P.Y.).

## Author contributions

P.Y. launched the study and obtained the funding. P.Y. and O.H. conceived the ideas, designed the experiments, analyzed the data, and wrote the manuscript. O.H., L.H.X., and K.V.S. carried out the experimental work. M.Z., D.H.M., F.J., J.H., J.W., E.K., and C.P. provided technical assistance and conceptual feedback or contributed to experimental work. All the authors discussed the results and had the opportunity to comment on the manuscript.

## Competing interests

The authors declare no competing interests.
