## [Peer Review File · Nature Communications]

Secondary structures that regulate mRNA translation provide insights for ASO-mediated modulation of cardiac hypertrophyREVIEWER COMMENTS

Reviewer #1 (Remarks to the Author):

In this study, Hedaya et al. set out to better understand the mechanisms regulating upstream open reading frame (uORF) activity and their inhibitory effect on translation of the corresponding main mRNA open reading frame (mORF). Based on the high degree of secondary structure within 5'UTRs, the authors created synthetic uORF constructs paired with dsRNA structure to examine their relationship to translational activity. These experiments indicated a dsRNA hairpin positioned just downstream of the uORF's AUG inhibits translation from the mORF and this effect can be attenuated by including mismatches within the stem loop. This effect of uORF-dsRNA was likely mediated by translation of the uORF from evidence including sedimentation with 80S ribosome. The authors next focused on the naturally occurring GATA4-uORF and confirmed its dsRNA structure through modeling and biochemical methods. Disrupting the GATA4 uORF stem loop, or mutating the uORF start codon both resulted in increased mORF translation. Importantly, this mechanism applied to additional genes with dsRNA-uORF structures. By designing antisense oligonucleotides (ASO) that bind to a uORF and either mimic the dsRNA structure (ASO1) or prevent dsRNA formation (ASO2) the authors were able to inhibit or promote mORF translation, respectively. This effect was consistent in hESC derived cardiomyocytes transfected with GATA4 uORF targeting ASO2 which elevated GATA4 levels and resulted in a hypertrophic phenotype, while transfection with ASO1 led to reduced cell size. Furthermore, in vivo administration of nanoparticle bound ASO1 reduced hypertrophy phenotypes in both TAC and ISO mouse models. The authors also optimized the ASO by introducing varying chemical modification to improve their use therapeutically, and demonstrated that they can also be applied to target mORF translation directly.

Overall, this manuscript reveals a novel mechanistic relationship linking uORF secondary structure with mORF regulation. Importantly, these mechanisms can be exploited with oligonucleotides to therapeutically modulate translation. It is refreshing to see a manuscript mine large datasets and apply the findings experimentally to a mechanistic model. While elegantly designed, implemented, and communicated, this study still needs some revisions in order to be suitable for publication, which are detailed below.

- 1) How would these ASO function with complex UTRs with multiple uORFs?
- 2) What is the specificity/biodistribution of nanoparticles and ASO bioavailability?
- 3) Shannon entropy is on the higher side for GATA4, so is it a good predictor of uORF structure?
- 4) Figure S2D left. It appears wt and Δ uORF are switched on graph.

Reviewer #2 (Remarks to the Author):

This study nicely shows how upstream ORFs and their correspondent dsRNA structure can be leveraged to modulate the expression of main ORFs. The authors show the efficacy of ASO treatments targeting dsRNAs in modulating cellular phenotypes induced by downstream encoded proteins (such as GATA4) both in cultured cells and in vivo. Overall, this is thorough, impactful study. I have the following comments:

1. The significance of the study is reduced by failure in demonstrating that upstream ORFs and their dsRNAs are indeed used by cells to modulate pathophysiological processes. The authors speculate in

the discussion that perhaps DDX3x could be acting on such structures to modulate the translation of downstream proteins. Is it possible to show that indeed this or other RNA binding proteins are differentially bound to upstream ORF structures or that the dsRNA structures themselves are disrupted upon stress?

2. For in vivo assessment of ASO efficacy in changing GATA4 levels, baseline or sham hearts should be used, because the expression of GATA4 can be sensitive to hypertrophic stress and following injury the authors are measuring phenotypically different hearts (Fig. 5).

3. When ASOs are systematically administered via tail-vein injection into adult mice, could you provide evidence that ASO are going inside the cardiomyocytes? Also, did you observe any side effects from the ASO treatments?

Minor:

4. AC16 cells are proliferating, feature that complicates interpretation of data on cardiomyocyte size and speculation on pro- or anti-hypertrophic effects (Fig. S3C). This is likely true for ESC-derived cardiomyocytes as well (Fig. 4). Can the authors corroborate their claim using hypertrophic, differentiation, maturation, and/or proliferation markers, to better understand the cellular phenotypes consequent to their GATA4 manipulation? Is there a reporter to actually measure GATA4 activity and quantitate the functional consequence of the observed protein level changes for this transcription factor?

5. In the methods section it is mentioned that "Age-matched male and female WT C57BL/6J mice were subjected to Sham or TAC surgery at 8-12 weeks of age." However, no sham data is presented.

Reviewer #3 (Remarks to the Author):

Recommendation: accept after major revisions

Though at one level, rather derivative, the manuscript by Hedaya et.al. asks an interesting, previously unasked question and provides sufficient new insights to justify publication in Nature Communications.

The question is do TIEs and uORfs work cooperatively to alter translational activity and, if so, what is the mechanism. Additionally, the authors attempt to extend work from the Crooke lab that suggested that PS ASOs could be designed to enhance translation of some m-RNAs by binding to uORFs or disrupting TIEs. The answer to the first question is yes and the data presented are reasonably compelling. The answer to second question is less compellingly made principally because I am concerned about the reproducibility of the supporting data. The authors also show that PS ASOs can be broken into two groups, one that inhibits translation of some m-RNAs and one that enhances translation. Here the authors do not provide an explanation that is clear to me.

Concerns:

Figure 2.

Having worked with shape reagents and having found them extremely variable. I would like to know how many times this experiment was repeated and see at least two replicates. Ideally, I would like to other proof of structure such as sensitivity to single strand nucleases or tiling with RNase H1 ASOs. In all experiments I would like to see evidence of the purity of the in vitro transcribed RNAs used as

substrates for translation.

Figure S1

The polysome profiles are extremely unconvincing and critical. I want to see multiple replicates here and in other figures with polysome profiles. I would like means and standard errors as these profiles are key the author's mechanistic explanation. Similarly, fig, 3 G needs to be reproduced and proof of reproducibility presented.

ASO designs

This topic was explored in both Liang et.al. papers and to some extent the current results replicate those results. In the discussion in the u ORF paper by Liang et.al., a plausible explanation was posited that the helicase activity necessary to eventually remove the ASO during scanning is acutely sensitive to Tm and to the nature of the 2' modifications used as well as the placement of the modifications. All of these elements are sensitive to RNA and

ASO sequences , the nature of the duplexes formed , the length of the ASO and the strength of the uORF , mORF, and nearby structures , e.g. TIEs. These topics should be discussed.

Reviewer #1:

In this study, Hedaya et al. set out to better understand the mechanisms regulating upstream open reading frame (uORF) activity and their inhibitory effect on translation of the corresponding main mRNA open reading frame (mORF). Based on the high degree of secondary structure within 5' UTRs, the authors created synthetic uORF constructs paired with dsRNA structure to examine their relationship to translational activity. These experiments indicated a dsRNA hairpin positioned just downstream of the uORF's AUG inhibits translation from the mORF and this effect can be attenuated by including mismatches within the stem loop. This effect of uORF-dsRNA was likely mediated by translation of the uORF from evidence including sedimentation with 80S ribosome. The authors next focused on the naturally occurring GATA4-uORF and confirmed its dsRNA structure through modeling and biochemical methods. Disrupting the GATA4 uORF stem loop, or mutating the uORF start codon both resulted in increased mORF translation. Importantly, this mechanism applied to additional genes with dsRNA-uORF structures. By designing antisense oligonucleotides (ASO) that bind to a uORF and either mimic the dsRNA structure (ASO1) or prevent dsRNA formation (ASO2) the authors were able to inhibit or promote mORF translation, respectively. This effect was consistent in hESC derived cardiomyocytes transfected with GATA4 uORF targeting ASO2 which elevated GATA4 levels and resulted in a hypertrophic phenotype, while transfection with ASO1 led to reduced cell size. Furthermore, in vivo administration of nanoparticle bound ASO1 reduced hypertrophy phenotypes in both TAC and ISO mouse models. The authors also optimized the ASO by introducing varying chemical modification to improve their use therapeutically, and demonstrated that they can also be applied to target mORF translation directly.

Overall, this manuscript reveals a novel mechanistic relationship linking uORF secondary structure with mORF regulation. Importantly, these mechanisms can be exploited with oligonucleotides to therapeutically modulate translation. It is refreshing to see a manuscript mine large datasets and apply the findings experimentally to a mechanistic model. While elegantly designed, implemented, and communicated, this study still needs some revisions in order to be suitable for publication, which are detailed below.

1) How would these ASO function with complex UTRs with multiple uORFs?

Response: Thanks for the reviewer's insightful questions. All the changes and updates based on reviewers' questions or suggestions were highlighted in blue in the manuscript text and figure legends.

To address this question, we will need to understand the relationship between the two uORFs (or even more than two uORFs). uORFs can work together in synergy (e.g., ATF4) or independently (e.g., MEF2C in our manuscript). Once the relationship between the uORFs is elucidated, the desired protein expression changes can be achieved similarly to how we showed for GATA4, which only requires a dsRNA downstream of the uORF to be targeted by ASO.

In our updated manuscript, we discovered that *MEF2C* mRNA 5' UTR contained two uORFs (uORF1 and uORF2), as shown in Supplementary Fig. 6e. We tested the Class I ASO targeting *MEF2C* uORF1 and uORF2. The results showed a similar mechanism of action as Class I ASO targeting GATA4 5' UTR that contains a single uORF. uORF2-specific ASO showed more robust inhibition of MEF2C protein expression than uORF1-specific ASO. We assume the effect could be case-by-case dependent on the number and location of multiple uORFs and their relationship in a given mRNA. Also, some uORFs may not be inhibitory against mORF translation. Therefore, elucidating the Class I and II ASOs in more complex 5' UTRs needs to be further investigated in many different mRNAs that contain one or more than one uORFs in the future.

2) What is the specificity/biodistribution of nanoparticles and ASO bioavailability?

Response: The nanoparticle In Vivo Transfection Reagent was obtained from Altogen Biosystems company (Cat. No. 5031; <https://altogen.com/product/nanoparticle-in-vivo-transfection-reagent/>). It can efficiently deliver small RNA systematically to multiple organs, including the heart, lungs, liver, pancreas, and kidney. To confirm the delivery of ASO1 in CMs, hearts, and potentially other organs, quantitative measurement using splint ligation-dependent qPCR assay was performed. Multiple organ tissues and CMs were isolated from WT mice 2 days post-ASO injection. The data suggested that ASO1 was efficiently delivered to the heart and CMs and also in multiple other organs, including the liver, kidney, and lung, but not to the brain or within the serum (Supplementary Fig. 5h). In our future work, we plan to develop further a new method for the heart- or cardiomyocyte-specific ASO delivery. However, this is beyond the scope of our work in this manuscript.

3) Shannon entropy is on the higher side for GATA4, so is it a good predictor of uORF structure?

Response: Low Shannon entropy is known from prior work as a feature that identifies functional structures in RNA. It is estimated, however, using RNA secondary structure prediction (specifically the prediction of pair probabilities using a partition function) and is, therefore, subject to inaccuracy in its estimation. The body of evidence supports that GATA4 has a dsRNA that is needed for the uORF; likely, the higher than-average Shannon entropy indicates a poor secondary structure prediction. However, we can still obtain the predicted structure for *GATA4* 5' UTR and validate its structure using SHAPE assay and dual luciferase reporter assay with mutagenesis for structural disruption. This effective integration of Shannon entropy prediction of *GATA4* uORF structure and experimental validations can complement the relatively poor prediction of uORF structure for *GATA4*.

We screened for uORF-dsRNA-bearing mRNAs by Shannon entropy (Supplementary Fig. 2f) by focusing on sequences with the lowest Shannon entropies. Our experiments (Supplementary Figs. 2g, 2h) support the requirement for these dsRNA structures and also support the strategy.

To better explain our logic, we added text and references to the Supplemental Methods that cite the precedent for using Shannon entropy. We also revised the main text to emphasize we examined only the lowest Shannon entropy sequences in our screen.

4) Figure S2D left. It appears wt and Δ uORF are switched on graph.

Response: We apologize for the confusion. The labeling of WT and Δ uORF is correct on the graph (in Supplementary Fig. 2e). To avoid confusion, we clarified the difference between the bar graph and western blot data in the figure legend. The left panel is firefly luciferase reporter activity of the *FLuc* mRNA containing WT or Δ uORF 5' UTR, which is correlated to FLuc mORF translation. The right panel is western blotting data of FLAG-tagged uORF-encoded peptides. The FLuc activity (indicating mORF translation) is anti-correlated relative to the uORF peptide expression.

Reviewer #2:

This study nicely shows how upstream ORFs and their correspondent dsRNA structure can be leveraged to modulate the expression of main ORFs. The authors show the efficacy of ASO treatments targeting dsRNAs in modulating cellular phenotypes induced by downstream encoded proteins (such as *GATA4*) both in cultured cells and in vivo. Overall, this is thorough, impactful study. I have the following comments:

1. The significance of the study is reduced by failure in demonstrating that upstream ORFs and their dsRNAs are indeed used by cells to modulate pathophysiological processes. The authors speculate in the discussion that perhaps DDX3x could be acting on such structures to modulate the translation of downstream proteins. Is it possible to show that indeed this or other RNA binding proteins are differentially bound to upstream ORF structures or that the dsRNA structures themselves are disrupted upon stress?

Response: Thanks for the reviewer's constructive questions. All the changes and updates based on reviewers' questions or suggestions were highlighted in blue in the manuscript text and figure legends.

We added new experimental data to support the role of DDX3X in regulating dsRNA-uORF activity in *GATA4* mRNA (see new Supplementary Figs. 2i-k). In brief, we knocked down a number of RNA helicases or translation factors (known to affect uORF-mORF translation balance) to test the effect on WT and Δ uORF *GATA4* 5' UTR (Supplementary Figs. 2i, j). We found that DDX3X exhibited a uORF-dependant regulation (i.e., knockdown only significantly affects uORF-containing 5' UTR). We hypothesized that this effect is dsRNA-dependent, so we tested the effect of DDX3X knockdown on our 5' UTR with dsRNA mismatches (MM) and showed that this mutation renders the 5' UTR insensitive to DDX3X expression (Supplementary Figs. 2k).

We stressed HEK293T cells with LPS, a pro-inflammatory stress condition that has been shown to induce DDX3X. We obtained similar results (Supplementary Figs. 2l). We found that mORF translation is enhanced, possibly driven by inhibited uORF activity and translation (Supplementary Figs. 2m, n). This effect was abolished when we knockdown DDX3X using siRNA.

Using RNA-binding protein immunoprecipitation, we tested whether the WT and the MM variant *GATA4* 5' UTRs are differentially bound to DDX3X, but found them to bind similarly (Supplementary Figs. 2o, p). It is possibly a reflection of the promiscuity of DDX3X, where the helicase is bound to RNA nonspecifically, but is only active when encountering a specific structure.

2. For in vivo assessment of ASO efficacy in changing *GATA4* levels, baseline or sham hearts should be used, because the expression of *GATA4* can be sensitive to hypertrophic stress and following injury the authors are measuring phenotypically different hearts (Fig. 5).

Response: We added sham controls and TAC samples for side-by-side comparisons. Supplementary Table 4 and Fig. 5 were updated, including Figs. 5b-h.

3. When ASOs are systematically administered via tail-vein injection into adult mice, could you provide evidence that ASO are going inside the cardiomyocytes? Also, did you observe any side effects from the ASO treatments?

Response: The nanoparticle In Vivo Transfection Reagent was obtained from Altogen Biosystems company (Cat. No. 5031; <https://altogen.com/product/nanoparticle-in-vivo-transfection-reagent/>). It can efficiently deliver small RNA systematically to multiple organs, including the heart, lungs, liver, pancreas, and kidney. To confirm the delivery of ASO1 in CMs, hearts, and potentially other organs, quantitative measurement using splint ligation-dependent qPCR assay was performed. Multiple organ tissues and CMs were isolated from WT mice 2 days post-ASO injection. The data suggested that ASO1 was efficiently delivered to the heart and CMs and in multiple other organs, including the liver, kidney, and lung, but not to the brain or within the serum (see new Supplementary Fig. 5h).

Because the liver was the predominant organ to uptake the ASO, we examined the liver-specific or systematic toxicity by measuring alanine transaminase (ALT) activity in mouse serum samples (see new Supplementary Fig. 5i). The results suggest no apparent organ toxicity of GATA ASO1 compared to control ASO. Consistent with no notable systematic toxicity, we did not observe significant changes in body weight, tibia length, and heart rate upon treatment of GATA4 ASO1 compared to control ASO.

Minor:

4. AC16 cells are proliferating, feature that complicates interpretation of data on cardiomyocyte size and speculation on pro- or anti-hypertrophic effects (Fig. S3C). This is likely true for ESC-derived cardiomyocytes as well (Fig. 4). Can the authors corroborate their claim using hypertrophic, differentiation, maturation, and/or proliferation markers, to better understand the cellular phenotypes consequent to their GATA4 manipulation? Is there a reporter to actually measure GATA4 activity and quantitate the functional consequence of the observed protein level changes for this transcription factor?

Response: We appreciate the reviewer's helpful suggestions for improving the work. We would like to clarify that our ASO treatments are carried out on fully differentiated ESC-

derived CMs, in which the number of cells seeded is the same. Therefore, any prior differentiation steps should be unaffected.

To address the other concerns, we added Figs. 4d and 4e to show that GATA4 target gene *Myh6* was increased in expression in homozygous Δ uORF hESC-derived CMs and was either decreased or increased by treatment of ASO1 and ASO2, respectively. These results suggest that uORF-mediated regulation of GATA4 translation can affect endogenous downstream GATA4 target gene expression. We are unable to clone the MYH6 or NPPA promoter effectively due to technical challenges and will need time to optimize and validate it using the artificial luciferase reporter system for the future. Moreover, we added Supplementary Fig. 4d to show that cell proliferation was not affected by ASO1 or ASO2 in AC16 human cardiomyocyte cell line or hESC-derived cardiomyocytes in culture.

5. In the methods section it is mentioned that "Age-matched male and female WT C57BL/6J mice were subjected to Sham or TAC surgery at 8-12 weeks of age." However, no sham data is presented.

Response: We added sham controls and TAC samples for side-by-side comparisons. Supplementary Table 4 and Fig. 5 were updated, including Figs. 5b-h.

Reviewer #3:

Though at one level, rather derivative, the manuscript by Hedaya et.al. asks an interesting, previously unasked question and provides sufficient new insights to justify publication in Nature Communications.

The question is do TIEs and uORFs work cooperatively to alter translational activity and, if so, what is the mechanism. Additionally, the authors attempt to extend work from the Crooke lab that suggested that PS ASOs could be designed to enhance translation of some mRNAs by binding to uORFs or disrupting TIEs. The answer to the first question is yes and the data presented are reasonably compelling. The answer to second question is less compellingly made principally because I am concerned about the reproducibility of the supporting data. The authors also show that PS ASOs can be broken into two groups, one that inhibits translation of some mRNAs and one that enhances translation. Here the authors do not provide an explanation that is clear to me.

Response: Thanks for the reviewer’s thoughtful comments and suggestions. All the changes and updates based on reviewers’ questions or suggestions were highlighted in blue in the manuscript text and figure legends.

We have provided or included the biological replicate data to demonstrate the reproducibility of our results shown in Fig. 2b, c, Fig. 3g, and Supplementary Fig. 1d (please see the detailed description below). These data, combined with new data for partially addressing the potential role of the trans-acting factor DDX3X in regulating GATA4 uORF-mORF translation, provide more substantial evidence to support the idea of cooperation between uORF and dsRNA (or TIE, translation inhibitory element, as Reviewer 3 referred to) to alter translation activity. In addition, we added more discussion to explain the two categories of ASOs used for inhibiting or enhancing translation (please see the answer to reviewer 3’s last question).

Concerns:

Figure 2.

Having worked with shape reagents and having found them extremely variable. I would like to know how many time this experiment was repeated and see at least two replicates. Ideally, I would like to SEE other proof of structure such as sensitivity to single strand nucleases or tiling with RNase H1 ASOs. In all experiments I would like to see evidence of the purity of the in vitro transcribed RNAs used as substrates for translation.

Response: Regarding the reproducibility concerns, we have two replicated SHAPE experiment data, and both are included in Fig. 2c (two replicated data were plotted) and Supplementary Fig. 2d (show the reproducibility of the two independent experiments). Notably, we have performed SHAPE assays in Fig. 2b, c (WT RNA with two different region-specific primers to detect WT RNA structure), Supplementary Fig. 2c (WT and mismatch mutant RNA that weakens the dsRNA structure in comparison with WT RNA), and

Supplementary Fig. 3a (ASO1 and ASO2 added in the reaction to strengthen and weaken the dsRNA structure, respectively). Also, we performed a mutagenesis-based dual luciferase reporter assay in Fig. 2e to confirm the effect of the mutation-based loss-of-function of RNA secondary structure at the GATA4 uORF region. Altogether, we used multiple independent and complementary approaches to validate the molecular mechanism driven by the uORF-dsRNA element for regulating mORF protein expression.

We repeated this SHAPE experiment (Fig. 2b) twice, and the second replicate data is shown in Figure A (SHAPE replicate data). Also, two replicated data using NAI were plotted in Fig. 2c.

To address the valid concern that different SHAPE reagents may show different results, we used NMIA to repeat the SHAPE experiment in parallel with NAI. We found that both NAI and NMIA SHAPE assay results mostly agree (see Figure B).

Figure B. Pearson correlation analysis of SHAPE assays using NMIA and NAI

We showed an agarose gel image of purified in vitro transcribed RNAs for the in vitro translation assay in Supplementary Fig. 1c.

We hope the new data, information, and clarification will suffice.

Figure S1.

The polysome profiles are extremely unconvincing and critical. I want to see multiple replicates here and in other figures with polysome profiles. I would like means and standard errors as these profiles are key the author's mechanistic explanation. Similarly, fig. 3G needs to be reproduced and proof of reproducibility presented.

Response: We are sorry for the confusion. In the initially submitted manuscript, the data was a representative result of 2 repeats. We experimented with a third round and incorporated all 3 replicates in the new Supplementary Fig. 1d (N=3 biological triplicates). Data are represented as mean \pm SD. In Supplementary Fig. 3c, we added the other two biological replicates data (replicate #2 and #3) in addition to the representative data (replicate #1) shown in Fig. 3g.

ASO designs

This topic was explored in both Liang et.al. papers and to some extent the current results replicate those results. In the discussion in the uORF paper by Liang et.al., a plausible explanation was posited that the helicase activity necessary to eventually remove the ASO during scanning is acutely sensitive to T_m and to the nature of the 2' modifications used as well as the placement of the modifications. All of these elements are sensitive to RNA and ASO sequences, the nature of the duplexes formed, the length of the ASO and the strength of the uORF, mORF, and nearby structures, e.g. TIEs. These topics should be discussed.

Response: We appreciate the suggestion from the reviewer and have added and modified the discussions in our manuscript as follows (highlighted in blue).

“This work offers a rational approach for developing translation-manipulating ASOs. Our understanding of uORF-dsRNA synergy rules allows the design of two classes of uORF-ASOs (**Figs. 1, 3**). The Class I uORF-ASOs that form a more stable artificial intermolecular dsRNA with a region immediately downstream of a uORF of a target mRNA recapitulate our observed uORF-dsRNA relationships and suppress the mORF translation (**Fig. 3, Supplementary Fig. 3a, 6f**). The PIC may pause upstream of the dsRNA with extended dwelling time and allow more efficient recognition of uORF start codon and joining of 60S ribosome large subunit, therefore promoting the translation of uORF and inhibiting mORF translation. In this dynamic process, as the ASO-mRNA duplex forms immediately downstream of the uORF AUG start codon in *GATA4* mRNA 5' UTR, the PIC likely unwinds the ASO-mRNA intermolecular duplex, possibly using an intrinsic component factor, namely, the RNA helicase eIF4A within the PIC⁵³, together with DDX3X^{11,38} (**Supplementary Fig. 2i-n**). This allows the 80S ribosome to start the translation elongation on the uORF sequence. Presumably, this potential helicase unwinding process during PIC scanning should be acutely sensitive to melting temperature (T_m), the nature of the 2' modifications used, and the placement of the modifications^{41,54}. This is supported by our finding that 2'-O-methyl plus four LNA nucleotides at the 3'-end of the ASO and LNA-based ASO show stronger translation repressive activity compared to 2'-O-methyl-, MOE-, or PS-based ASO for *GATA4* ASO1 (**Fig. 6a-d**). Furthermore, chemical modifications and their positions may be sensitive to target mRNA and ASO sequences, the nature of the mRNA-ASO duplexes formed, the

length of the ASO, the strength of the uORF and mORF (e.g., Kozak sequence context), and nearby structures such as dsRNA or other translation inhibitory elements^{41,54}.

In contrast, the Class II uORF-ASOs that disrupt the endogenous dsRNA downstream of, and adjacent to, a uORF, recapitulate our dsRNA mismatch mutations or DDX3X-mediated dsRNA unwinding and alleviate uORF suppression of the mORF (**Fig. 3, Supplementary Fig. 3a, 6f**). In this case, the PIC may skip the uORF start codon and reach the mORF start codon to assemble into an 80S ribosome to translate the primary CDS and synthesize the main protein product. Our other class of ASOs, Class II, requires the presence of a dsRNA downstream of the uORF start codon (**Fig. 3, Supplementary Fig. 6f**). We characterized the dsRNA within the *GATA4* 5' UTR and showed its disruption via ASOs could increase GATA4 protein levels (**Figs. 3, 4**). Other studies have demonstrated similar effects by disrupting dsRNA to boost protein expression^{16,41}. However, it is unclear how exactly these targeted dsRNAs are inhibitory in these cases. Our work has provided insights into the mechanism of action for uORF-dsRNA-mediated translational control and the application of our “distance rule” to develop mechanism-based ASOs. Class I and Class II ASOs can be utilized to control protein levels for various applications, and more studies are needed to elucidate the dichotomy between these two targeting mechanisms.”

REVIEWERS' COMMENTS

Reviewer #1 (Remarks to the Author):

The authors have addressed my concerns.

Reviewer #2 (Remarks to the Author):

We thank the authors for addressing our concerns and adding important new data to reinforce their conclusion.

Reviewer #3 (Remarks to the Author):

The authors have responded to my concerns and the manuscript is acceptable for publication.